# ◎ SPIRAL: Semantic-Aware Progressive LiDAR Scene Generation and Understanding

**Dekai Zhu**[1,4,*] **Yixuan Hu**[1,*] **Youquan Liu**[2] **Dongyue Lu**[3]
**Lingdong Kong**[3] **Slobodan Ilic**[1]

[1]Technical University of Munich    [2]Fudan University
[3]National University of Singapore    [4]Munich Center for Machine Learning

https://dekai21.github.io/SPIRAL/

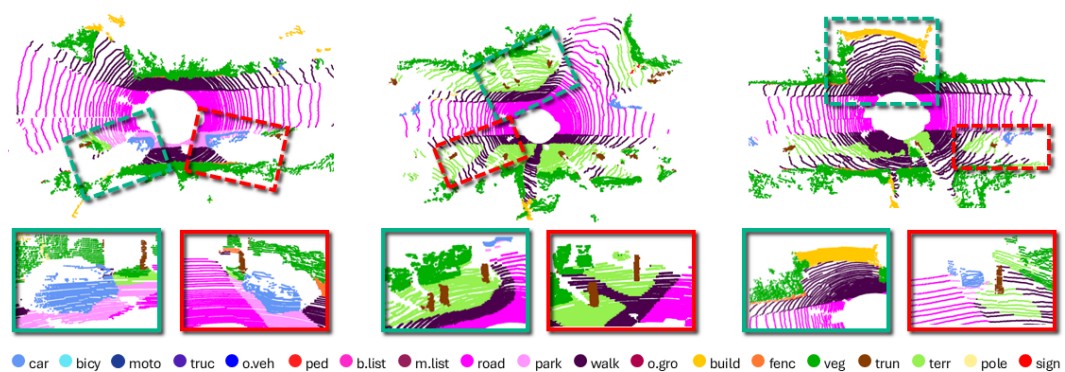

Figure 1: **Visualization of LiDAR scenes and their semantic labels jointly generated by SPIRAL,** exhibiting high geometric fidelity and semantic–geometric consistency.

## Abstract

Leveraging recent diffusion models, LiDAR-based large-scale 3D scene generation has achieved great success. While recent voxel-based approaches can generate both geometric structures and semantic labels, existing range-view methods are limited to producing unlabeled LiDAR scenes. Relying on pretrained segmentation models to predict the semantic maps often results in suboptimal cross-modal consistency. To address this limitation while preserving the advantages of range-view representations, such as computational efficiency and simplified network design, we propose SPIRAL, a novel range-view LiDAR diffusion model that simultaneously generates depth, reflectance images, and semantic maps. Furthermore, we introduce novel semantic-aware metrics to evaluate the quality of the generated labeled range-view data. Experiments on the SemanticKITTI and nuScenes datasets demonstrate that SPIRAL achieves state-of-the-art performance with the smallest parameter size, outperforming two-step methods that combine the generative and segmentation models. Additionally, we validate that range images generated by SPIRAL can be effectively used for synthetic data augmentation in the downstream segmentation training, significantly reducing the labeling effort on LiDAR data.

## 1 Introduction

By providing accurate distance measurements regardless of ambient illumination, LiDAR plays a crucial role in scene understanding and navigation for robotics and autonomous driving [11, 60, 10, 13,

---

*Equal contributions.

39th Conference on Neural Information Processing Systems (NeurIPS 2025).

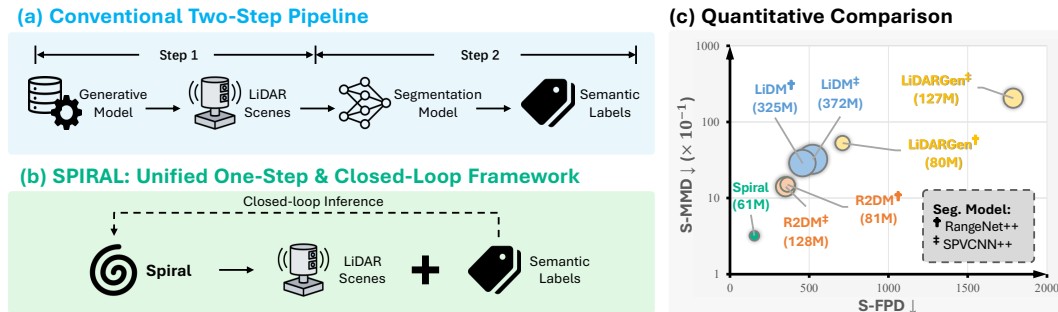

Figure 2: (a) **Two-step methods**: Existing range-view LiDAR generative models typically generate only depth and reflectance images, requiring an additional pre-trained segmentation model to predict semantic labels. (b) **SPIRAL**: In contrast, Spiral jointly generates depth, reflectance, and semantic maps. A closed-loop inference mechanism (highlighted in the dash arrow) further improves cross-modal consistency. (c) **Results**: Spiral achieves state-of-the-art performance with the smallest parameter size (61M) among the related methods.

32, 26, 27, 58]. However, collecting and annotating large-scale LiDAR datasets is both expensive and time-consuming [12, 64, 41, 38, 63]. To address this issue, recent research has increasingly focused on using denoising diffusion probabilistic models (DDPMs) [14] for LiDAR generative modeling, aiming to create tools capable of generating unlimited LiDAR scenes [73, 48, 16, 42, 43, 37, 34].

Existing generative approaches can be categorized into voxel-based methods [43, 29, 49, 4] and range-view-based methods [73, 42, 48, 16]. The former divides the 3D space into regular volumetric grids (*i.e.*, voxels) and captures detailed geometric structures with 3D convolutional networks [8]. However, they often suffer from high memory consumption and computational overhead [52]. Methods based on range-view, on the other hand, project LiDAR point clouds onto a 2D cylindrical image plane using the sensor azimuth and elevation angles [73, 48, 16, 33]. This leads to compact 2D representations of depth and reflectance, allowing for efficient processing via 2D convolutional networks [1, 23]. These methods are significantly more memory-efficient and computationally lightweight [62, 61].

In this work, we aim to address two limitations in existing range-view generative methods:

**1.** While recent models such as LiDARGen [73] and R2DM [42] generate high-fidelity LiDAR scenes, their outputs are restricted to depth and reflectance images, without producing semantic labels.

**2.** Existing evaluations extract global features for each scene from three perspectives (range-view image, Cartesian point cloud, and BEV projection) to assess the distributional similarity between generated and real scenes. However, none of these evaluation methods consider semantic labels.

For the first limitation, a straightforward solution is to adopt a two-step pipeline: first, produce unlabeled LiDAR scenes using generative models [73, 42, 48], and then apply a pretrained segmentation model (*e.g.*, RangeNet++ [40]) to predict the corresponding semantic labels, as depicted in Figure 2 (a). However, this approach often results in suboptimal performance due to two key issues: (1) The generative and segmentation models are trained independently, which hinders shared representations between the two tasks and reduces training efficiency. (2) The semantic maps, being predicted post hoc, cannot serve as conditional guidance during generation, leading to limited consistency between semantics and other modalities, including depth and reflectance.

Therefore, we propose a novel semantic-aware range-view LiDAR diffusion model, named **SPIRAL**, as depicted in Figure 2 (b), with the following key features:

• **Semantic-aware generation:** Our framework aims to jointly generate depth, reflectance, and semantic maps from the Gaussian noise, different from existing models that lack semantic awareness and require separate segmentation models to obtain semantic labels.

• **Progressive semantic prediction:** At each denoising step, **SPIRAL** outputs an intermediate semantic map which is aggregated via exponential moving averaging (EMA) to suppress noise and produce stable, per-pixel confidence scores. The EMA trace serves as both the final semantic output and the basis for the closed-loop inference.

• **Closed-loop inference:** Once the prediction confidence exceeds a threshold, the semantic map is fed back into the model as the condition to guide the generation of both depth and reflectance. By alternating between conditional and unconditional steps, SPIRAL enables joint refinement of geometry and semantics, thereby enhancing cross-modal consistency.

For the second limitation, we extend all three types of metrics with semantic awareness, enabling a comprehensive assessment of geometric, physical, and semantic quality in the generated LiDAR scenes. (1) For the evaluation in range-view image and Cartesian point cloud perspectives that rely on pretrained models such as RangeNet++ [40] and PointNet [47] to extract learning-based global features, we integrate the semantic map conditional module from LiDM [48], which is originally designed for semantic-to-LiDAR generation, to encode semantic labels. The encoded semantic features are then concatenated with the original global features to form semantic-aware global features. (2) For the evaluation in BEV projection perspective that produces rule-based global features using 2D histograms over the xy-plane, we compute 2D histograms for each semantic category individually and then concatenate them into a unified semantic-aware histogram representation.

Experiments on the SemanticKITTI [3] and nuScenes [5] datasets demonstrate that Spiral achieves state-of-the-art performance in labeled LiDAR scene generation with the smallest parameter size, as depicted in and Fig. 1 and Fig. 2 (c). Moreover, we demonstrate that Spiral-generated samples can be effectively used as synthetic data to augment downstream training, which is particularly valuable for autonomous driving tasks that require large-scale training data [39, 70, 71]. To summarize, the key contributions of this work are as follows:

- We propose a novel state-of-the-art semantic-aware range-view LiDAR diffusion model, SPIRAL, which jointly produces depth and reflectance images along with semantic labels.

- We introduce unified evaluation metrics that comprehensively evaluate the geometric, physical, and semantic quality of generated labeled LiDAR scenes.

- We demonstrate the effectiveness of the generated LiDAR scenes for training segmentation models, highlighting Spiral's potential for generative data augmentation.

## 2 Related Works

**LiDAR Generation from Range Images.** LiDARGen [73] pioneers diffusion-based LiDAR generation by learning a score function [18] that models the log-likelihood gradient in range-view image space. Building on this foundation, LiDM [48] advances conditional generative modeling, enabling synthesis from multiple input modalities. R2DM [42] enhances unconditional LiDAR scene generation through diffusion models and provides in-depth analysis of crucial components that improve generation fidelity. RangeLDM [16] optimizes the computational efficiency to achieve real-time generation, while Text2LiDAR [59] develops text-guided generation capabilities for semantic control. However, for semantic segmentation applications, these methods still require additional dedicated models, increasing computational cost.

**LiDAR Generation from Voxel Grids.** Several studies investigate the generation of complete LiDAR scenes in Cartesian space, aiming to preserve geometric reconstruction accuracy [29, 43, 4, 49, 30, 57, 67, 54, 56, 45]. SemCity [29] leverages a triplane representation to model outdoor LiDAR scenes by projecting the 3D space into three orthogonal 2D planes. XCube [49] utilizes a hierarchical voxel latent diffusion model to generate large-scale scenes. [43] proposes to generate semantic LiDAR scenes using the latent DDPM without relying on intermediate image projections or coarse-to-fine multi-resolution modeling. DynamicCity [4] utilizes a VAE [22] model and a DiT-based [44] DDPM to generate large-scale, high-quality dynamic 4D scenes.

**LiDAR Semantic Segmentation.** Various approaches are proposed for LiDAR scene segmentation from different representations, including range-view [40, 1, 23, 62, 25, 7], BEV [65], voxel [8, 72, 24, 15], and multi-view fusion [19, 20, 36, 6, 46, 39, 31, 35, 17]. As a representative range-view segmentation model, RangeNet++ [40] offers a real-time and efficient solution for LiDAR semantic segmentation. Cylinder3D [72] proposes a cylindrical representation that is particularly well-suited for outdoor scenes, addressing the irregularity and sparsity issues in LiDAR point clouds. SPVCNN [55] serves as a point-voxel fusion framework that integrates point cloud and voxel representations to leverage their complementary advantages for improved segmentation performance.

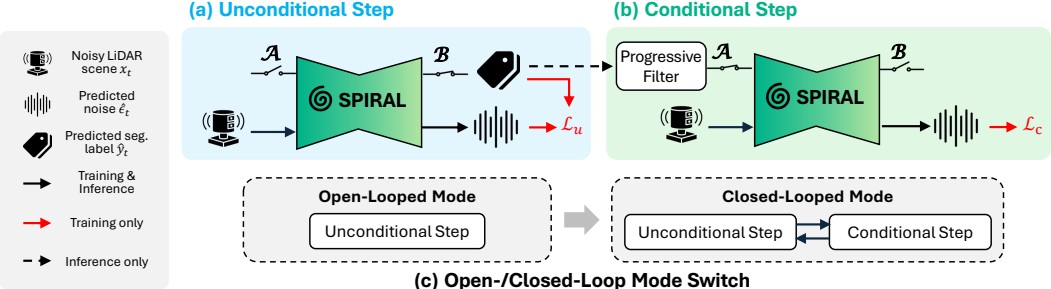

Figure 3: **(a) Unconditional Step:** Spiral takes noisy LiDAR scenes $x_t$ as input and predicts both the semantic map $\hat{y}_t$ and the noise $\hat{\epsilon}_t$, where the switch $\mathcal{A}$ is off and $\mathcal{B}$ is on. **(b) Conditional Step:** Spiral predicts $\hat{\epsilon}_t$ conditioned on the given semantic map $y$, where $\mathcal{A}$ is on and $\mathcal{B}$ is off. **(c)** During inference, Spiral begins in an **open-loop mode** with unconditional steps. Once the predicted semantic map smoothed by the progressive filter reaches high confidence, Spiral switches to a **closed-loop mode** that alternates between unconditional and conditional steps, enhancing cross–modal consistency.

# 3 Methodology

## 3.1 Preliminaries of Diffusion Models for Range-View LiDAR Generation

The diffusion models [14] generate data by simulating a stochastic process. Starting from a real sample $x_0 \sim q(x_0)$, a forward process gradually adds Gaussian noise over $T$ steps, such that the final sample approximates a standard normal distribution, ie $q(x_T) \approx \mathcal{N}(0, I)$. For range-view LiDAR generation, the sample is represented as $x_0 \in \mathbb{R}^{H \times W \times d}$, where $H$ and $W$ denote the height and width of the range image, and $d$ denotes the number of modalities. In LiDARGen [73] and R2DM [42], $d = 2$ as both depth and reflectance images are generated, while $d = 1$ in LiDM [48] since it only generates depth images. The model $\epsilon_\theta$ with parameters $\theta$ is trained to predict the noise $\epsilon$ added at an intermediate step $t \in \{1, \ldots, T\}$, by minimizing the following objective:

$$\mathcal{L} = \mathbb{E}_{t,x_0,\epsilon} \left[ \|\epsilon_\theta(x_t, t, a) - \epsilon\|_2^2 \right], \tag{1}$$

where $\epsilon \sim \mathcal{N}(0, I)$ is the random noise added during the forward process, and $a$ denotes the conditional input, such as textual description or semantic maps $y \in \mathbb{R}^{H \times W \times C}$, where $C$ denotes the number of categories. During inference, the model starts from a random noise sample $x_T \sim \mathcal{N}(0, I)$ and iteratively denoises it to generate a novel sample $x_0'$. At each step, an additional noise $\eta \sim \mathcal{N}(0, I)$ is added on $x_t$ to diversify the final results. Building upon the vanilla DDPM, a more flexible diffusion framework [50] is proposed by introducing a continuous time variable $t \in [0, 1]$, which enables flexible control over the number of sampling steps during inference, allowing a trade-off between generation speed and quality.

## 3.2 SPIRAL: Semantic-Aware Pregressive LiDAR Generation

As previously discussed, although existing range-view LiDAR generative models [73, 48, 42] have demonstrated impressive performance, they are limited to producing only depth and reflectance modalities. While LiDM [48] generates LiDAR scenes conditioned on semantic maps, it requires the semantic maps to be provided beforehand. Alternatively, two-step pipelines that first generate LiDAR scenes and then predict semantic labels suffer from low training efficiency and limited cross-modal consistency. Inspired by the insight that diffusion models can serve as powerful representation learners for various tasks such as classification and segmentation [2, 68, 28, 69], we propose a novel **s**emantic-aware **p**rogress**i**ve **ra**nge-view **L**iDAR diffusion model, named **SPIRAL**, as illustrated in Figure 2 (b), to address these limitations. Spiral contains three major innovations:

**Complete Semantic Awareness.** The semantic awareness of Spiral contains two aspects: **(1)** In addition to generating depth and reflectance images, Spiral also predicts the corresponding semantic maps; **(2)** Spiral enables conditional generation of depth and reflectance images guided by a given semantic map. It alternates between two types of steps: **unconditional** and **conditional**. To control the switching between them, we introduce two control switches, $\mathcal{A}$ and $\mathcal{B}$, as illustrated in Figure 3.

Switches $\mathcal{A}$ and $\mathcal{B}$ follow an exclusive-or (XOR) relationship. Their on/off states are as follows:

$$\begin{cases} \mathcal{A} = 0, \ \mathcal{B} = 1; & \text{for the unconditional step,} \\ \mathcal{A} = 1, \ \mathcal{B} = 0; & \text{for the conditional step,} \end{cases} \tag{2}$$

where 0 denotes the off state and 1 denotes the on state. During training, in the unconditional step, the Spiral with learnable parameters $\theta$, $\epsilon_\theta$, simultaneously predicts both the semantic map $\hat{y}_t$ and the noise $\hat{\epsilon}_t$ on the noisy LiDAR scene $x_t$, i.e.,

$$\hat{\epsilon}_t, \hat{y}_t \leftarrow \epsilon_\theta(x_t), \tag{3}$$

and the corresponding training loss $\mathcal{L}_u$ is calculated as:

$$\mathcal{L}_u = \mathcal{L}_{\text{noise}} + \mathcal{L}_{\text{sem}} = \text{MSE}(\epsilon_t, \hat{\epsilon}_t) + H(\hat{y}_t, y), \tag{4}$$

where $\text{MSE}(\cdot)$ denotes the mean squared error, and $H(\cdot)$ represents the cross-entropy loss. In the conditional step, $\epsilon_\theta$ takes the semantic map $y$ as conditional input and only predicts the denoising residual:

$$\hat{\epsilon}_t \leftarrow \epsilon_\theta(x_t, y), \tag{5}$$

with the training loss $\mathcal{L}_c$ calculated as:

$$\mathcal{L}_c = \mathcal{L}_{\text{noise}} = \text{MSE}(\epsilon_t, \hat{\epsilon}_t). \tag{6}$$

We use a random variable $\psi \sim \text{Uniform}(0, 1)$ to determine the mode for each training step. Therefore,

$$\mathcal{L} = \mathcal{L}_c \cdot \mathbb{I}(\psi \leq \psi_c) + \mathcal{L}_u \cdot \mathbb{I}(\psi > \psi_c), \tag{7}$$

where $\mathbb{I}(\cdot)$ is the indicator function and $\psi_c$ is the ratio of training the conditional step. Empirically, we set $\psi_c$ as 0.5 to balance the training of these two step types.

**Progressive Semantic Predictions.** During inference, Spiral predicts the semantic map $\hat{y}_t$ at each unconditional step. To mitigate the inherent stochasticity of the diffusion process and improve stability, we apply an exponential moving average (EMA) to obtain the smoothed predictions $\bar{y} \in \mathbb{R}^{H \times W \times C}$. The EMA is initialized as $\bar{y}_T = \hat{y}_T$, and updated recursively for $t = T-1$ to 0 as:

$$\bar{y}_t \leftarrow \alpha \cdot \hat{y}_t + (1 - \alpha) \cdot \bar{y}_{t+1}, \tag{8}$$

where $\alpha$ denotes the EMA smoothing factor. At the end of inference, Spiral outputs not only the depth and reflectance images, but also the final smoothed semantic prediction $\bar{y}_0$.

**Closed-Loop Inference.** Two-step methods suffer from limited cross-modality consistency, since the predicted semantic map post hoc cannot guide the previous generation. To address this issue, Spiral introduces a novel closed-loop inference mechanism, where the semantic predictions $\bar{y}_t$ are continuously fed back into the model as conditional inputs during inference. Notably, to alleviate the potential artifacts in $\bar{y}_t$ that could degrade the generation quality, Spiral employs a confidence-based filtering strategy to select the reliable predictions as conditions. Specifically, during inference from step $t = T$ to 0, Spiral starts with an open-loop mode by default. However, if more than the proportion $\delta$ of the pixels in $\bar{y}_t$ have confidence scores exceeding $\delta$, it switches to the closed-loop mode, as depicted in Figure 3 (c). For instance, setting $\delta$ to 0.8 requires that over 80% of the pixels in $\bar{y}_t$ have prediction confidence scores exceeding 0.8. Once this condition is met, Spiral enters an alternating loop between unconditional and conditional steps: **(1)** In the unconditional step, Spiral predicts both $\hat{\epsilon}_t$ and $\hat{y}_t$. **(2)** In the conditional step, Spiral predicts $\hat{\epsilon}_t$ conditioned on $\bar{y}_t$, thereby improving the consistency between $\bar{y}_0$ and $x_0$ eventually. A quantitative study on the benefits of the closed-loop mode and the effect of the confidence threshold $\delta$ is presented in the experimental section.

**Overall Architecture.** Spiral adopts a 4-layer Efficient U-Net [50] as the backbone and follows the default continuous DDPM framework. Spiral takes as input the perturbed depth and reflectance images $x_t$, along with semantic maps $y$ encoded as RGB images. Notably, in unconditional step, the semantic maps are replaced with zero padding to disable semantic guidance. On the output side, Spiral utilizes two heads to separately predict the diffusion residuals $\hat{\epsilon}_t$ and the segmentation labels $\hat{y}_t$. Each output branch consists of a 2D convolutional layer followed by a sequential MLP layer. More details about the architecture of Spiral are provided in the appendix.

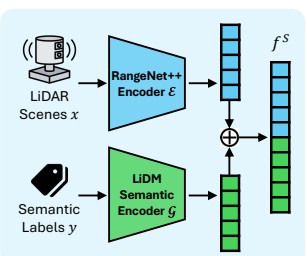
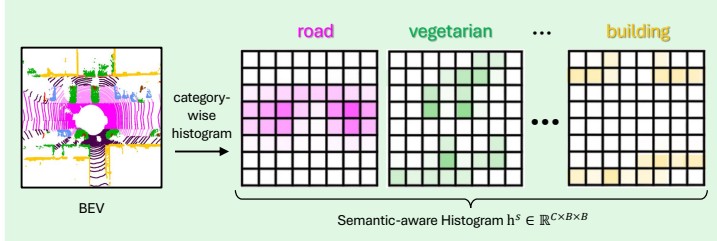

Figure 4: **(a) Range-view** based semantic-aware feature $f^s$ is constructed by concatenating the features extracted by the RangeNet++ [3] encoder and the LiDM [48] semantic encoder from the LiDAR scene $x$ and the semantic map $y$, respectively. **(b) BEV-based** semantic-aware feature $h^s$ is constructed by aggregating per-category 2D histograms.

## 3.3 Semantic-Aware Metrics for Better LiDAR Generation Evaluations

Recent range-view LiDAR studies [73, 42, 48] assess the quality of generated scenes from three perspectives: range-view images, Cartesian point clouds, and BEV projections. **(1)** For the range image- and point cloud-based evaluations, they rely on pretrained models such as RangeNet++ [40] and PointNet [47] to extract **learning-based features** $f$ for each scene. The extracted features from the real and generated sets are then used to compute the Fréchet Range Distance (FRD) [73], Fréchet Point Cloud Distance (FPD) [42], and Maximum Mean Discrepancy (MMD), which quantify the distributional similarity between these two sets. **(2)** For the BEV-based evaluation, **rule-based descriptions** $h$ are computed for each scene using 2D histograms over the xy-plane. These features, extracted from both the real and generated sets, are then used to calculate the MMD and Jensen–Shannon Divergence (JSD). However, none of these evaluation methods take semantic labels into account. Since the "ground truth" for semantic labels on the generated scene is unavailable, standard metrics like mIoU cannot be directly applied. Thereby, we propose to assess the quality of the generated labeled LiDAR scenes based on semantic-aware features $f^s$ and $h^s$.

**Learning-based Semantic Features.** For the range-view based evaluation, following LiDAR-Gen [73], we use the RangeNet++ [40] as encoder $\mathcal{E}$. Additionally, we propose to use a semantic map encoder $\mathcal{G}$ to extract the semantic latent features. For $\mathcal{G}$, we use the semantic conditional module in LiDM [48]. Given a LiDAR scene $x$ with semantic labels $y$, the semantic-aware features $f^s$ are obtained by the concatenation of these two features, as depicted in Figure 4 (a):

$$f^s \leftarrow \mathcal{E}(x) \oplus \mathcal{G}(y). \tag{9}$$

The extracted features from the real and generated sets, $\{f^s\}_r$ and $\{f^s\}_g$, are ultimately used to compute the semantic-aware Fréchet Range Distance (S-FRD) and semantic-aware Maximum Mean Discrepancy (S-MMD), extending FRD and MMD to incorporate semantic information. For the Cartesian-based evaluation, we adopt the same procedure to extract $f^s$, while the only difference is that the RangeNet++ [40] is replaced with PointNet [47]. Similarly, $\{f^s\}_r$ and $\{f^s\}_g$ are used to compute the semantic-aware Fréchet Point Cloud Distance (S-FPD) and S-MMD.

**Rule-based Semantic Features.** The BEV-based evaluation in [73, 42] divides the xy-plane into a grid of $B{\times}B$ bins and computes a 2D BEV histogram as the rule-based feature $h \in \mathbb{R}^{B \times B}$. However, the spatial distribution of points belonging to different categories is obviously distinct. For instance, "road" points are typically concentrated near the region along the x-axis, while "building" and "vegetation" points tend to appear farther away. The distribution of points across different categories within a scene encodes rich semantic information and effectively reflects the overall semantic structure of the scene. Thereby, we propose to compute histograms for each category individually and aggregate them into a semantic-aware histogram $h^s \in \mathbb{R}^{C \times B \times B}$, as depicted in Figure 4 (b). Hence, only when a generated and a real scene share both similar point distributions and semantic classifications can their histograms be similar. The extracted descriptions from the real and generated sets, $\{h^s\}_r$ and $\{h^s\}_g$, are used to compute the semantic-aware JSD (S-JSD) and S-MMD.

# 4 Experiments

## 4.1 Experimental Setup

**Datasets.** We conduct an extensive experimental study on SemanticKITTI [3] and nuScenes [5] datasets and follow their official data splits. SemanticKITTI contains 23k annotated LiDAR scenes with 19 semantic classes, while nuScenes contains 28k LiDAR scenes with 16 semantic classes. During pre-processing, the LiDAR scenes are projected into range-view images of spatial resolutions 64×1024 and 32×1024, respectively. To further assess robustness, we also evaluate Spiral-based generative data augmentation on the fog and wet-ground subsets of Robo3D [24], which simulate adverse weather conditions for out-of-distribution testing.

**Details on Training & Inference.** We train SPIRAL on NVIDIA A6000 GPUs with 48 GB VRAM for 300k steps using the Adam optimizer [21] with a learning rate of 1e-4. The training process takes $\sim$ 36 hours. For the generative models in two-step baseline methods, including LiDARGen [73], LiDM [48], and R2DM [42], we follow the official training settings. To obtain semantic labels for the generated unlabeled LiDAR scenes, we use RangeNet++ [40] with official pretrained weights and SPVCNN++ [36], an improved implementation of SPVCNN [55] provided in UniSeg [36] codebase. During inference, the number of function evaluations (NFE) [42], *i.e.*, the number of sampling steps, is set to 256 for both Spiral and R2DM. We also run LiDM using the DDIM [51] sampling method with 256 steps for fair comparison. LiDARGen models the denoising process with 232 noise levels and requires 5 steps per level by default, resulting in a total NFE of 1160. Following [73], we generate 10k samples per method for evaluation.

**Evaluation Metrics.** We follow previous works [73, 42] to assess the quality of generated LiDAR scenes from the perspectives of range images, point clouds, and BEV projections. Importantly, we report the evaluation of the generated labeled LiDAR scenes using our newly proposed semantic-aware metrics introduced in Section 3.3, including S-FRD, S-FPD, S-MMD, and S-JSD. For all the metrics, *lower* values indicate *better* generative quality. Note that the samples generated by LiDM [48] contain only depth images, which prevents their evaluation in range-view-based benchmarks where the RangeNet++ [16] encoder requires the reflectance channel as input.

## 4.2 Experimental Results

**Evaluation on SemanticKITTI.** We report the experimental results on the SemanticKITTI [3] dataset in Table 1. Despite having the smallest parameter size of only **61M**, Spiral achieves the best performance across all semantic-aware metrics, outperforming the two-step method, R2DM [42] & SPVCNN++ [36], by **31.03%**, **56.33%**, and **50.94%** on S-FRD, S-FPD, and S-JSD, respectively. For the previous metrics that evaluate only the unlabeled LiDAR scenes, Spiral outperforms R2DM on most metrics, indicating that the additional semantic prediction task does not compromise the generation quality of depth and reflectance images. Surprisingly, the more advanced segmentation model SPVCNN++ performs worse than RangeNet++ on the unlabeled scenes generated by LiDAR-Gen [73] and LiDM [48], resulting in inferior performance on semantic-aware metrics. We attribute this drop to the higher sensitivity of larger models to noise, compounded by the greater noise present in the LiDAR scenes generated by LiDARGen and LiDM. Although the performance of SPVCNN++ improves after jittering-based fine-tuning, it still lags behind RangeNet++. Further discussion of this issue is provided in the appendix. The generated labeled LiDAR scenes from Spiral and other baseline methods, as shown in Figure 5, demonstrate the superior performance of Spiral in both geometric and semantic aspects.

**Evaluation on nuScenes.** We report the experimental results on the nuScenes [5] dataset in Table 2. Spiral consistently outperforms the other baseline methods on all metrics with the smallest parameter size. Compared with the second best method (R2DM [42] & RangeNet++ [40]), Spiral achieves improvements of **49.03%**, **67.84%**, and **46.79%** on S-FRD, S-FPD, and S-JSD, respectively. Figure 6 presents the generated scenes from Spiral and the baseline methods for qualitative comparison, showing the superior performance of Spiral in both geometric and semantic aspects.

**Generative data augmentation.** We evaluate the effectiveness of using Spiral's generated samples to augment the training set for segmentation learning on SemanticKITTI [3]. Using SPVCNN++ [36] as the segmentation backbone, we compare Spiral with R2DM [42] & RangeNet++ [40] under different ratios of available real labeled data. As shown in the first row of Table 3, the generated samples

Table 1: **Comparisons with state-of-the-art LiDAR generation models** on the SemanticKITTI [3] dataset. We evaluate methods using the Range View, Cartesian, and BEV representations. Symbols † and ‡ denote the RangeNet++ [40] backbone and the SPVCNN++ [36] backbone, respectively. The parameter size includes both the generative and segmentation models. The **best** and second best scores under each metric are highlighted in **bold** and underline.

| Method | Param (M) | NFE | Range View | | | | Cartesian | | | | BEV | | | |
|---|---|---|---|---|---|---|---|---|---|---|---|---|---|---|
| | | | FRD↓ (×1) | MMD↓ (×10⁻²) | S-FRD↓ (×1) | S-MMD↓ (×10⁻²) | FPD↓ (×1) | MMD↓ (×10⁻¹) | S-FPD↓ (×1) | S-MMD↓ (×10⁻¹) | JSD↓ (×10⁻²) | MMD↓ (×10⁻³) | S-JSD↓ (×10⁻²) | S-MMD↓ (×10⁻³) |
| LiDARGen† [73] | 30+50 | 1160 | 681.37 | 47.87 | 1216.61 | 35.65 | 115.17 | 46.37 | 710.79 | 52.71 | 13.23 | 2.19 | 28.65 | 10.96 |
| LiDARGen‡ [73] | 30+97 | | | | 1708.05 | 61.42 | | | 1366.17 | 103.12 | | | 54.84 | 68.16 |
| LiDM† [48] | 275+50 | 256 | — | — | — | — | 108.70 | 33.87 | 458.33 | 28.60 | 4.56 | 0.29 | 16.69 | 5.59 |
| LiDM‡ [48] | 275+97 | | | | — | — | | | 522.47 | 32.39 | | | 21.03 | 6.23 |
| R2DM† [42] | 31+50 | 256 | 192.81 | 4.93 | 559.26 | 11.56 | 19.29 | 2.30 | 363.16 | 15.00 | **3.73** | 0.16 | 18.13 | 3.91 |
| R2DM‡ [42] | 31+97 | | | | 555.09 | 11.50 | | | 351.73 | 14.06 | | | 18.67 | 3.97 |
| SPIRAL (Ours) | 61 | 256 | **170.18** | **4.81** | **382.87** | **4.10** | **8.06** | **1.10** | **153.61** | **3.20** | 3.76 | **0.15** | **9.16** | **1.41** |

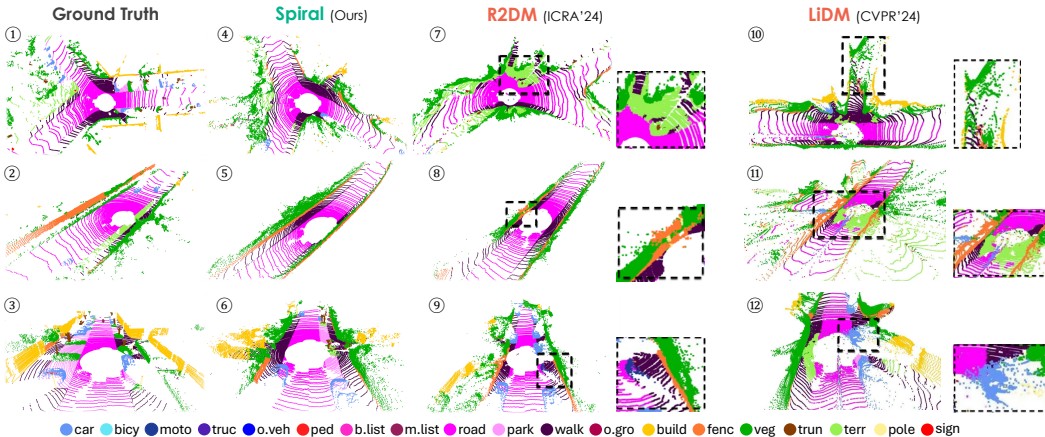

Figure 5: **Visualizations of generated LiDAR scenes** on SemanticKITTI [3]. For two-step methods, we use the labels produced by RangeNet++ [40] due to its superior performance over SPVCNN++ [36]. Artifacts are highlighted with dashed boxes. Examples of semantic artifacts are shown in ⑦, ⑧, ⑨, and ⑪, while geometric artifacts such as local distortion and large noise are illustrated in ⑩ and ⑫.

from Spiral consistently improve the performance of SPVCNN++ and outperform those from R2DM. Detailed per-category results are provided in the appendix. Additionally, we evaluate SPVCNN++ under the same settings on out-of-distribution subsets, fog and wet-ground, from Robo3D [24]. As shown in the second and third rows of Table 3, although Spiral is not fine-tuned for such extreme weather conditions, its generated data still enriches the training set and improves performance under these challenging scenarios.

### 4.3 Experimental Analysis

**Impact of the NFE.** Intuitively, increasing the number of function evaluations (NFE) improves the quality of the generated samples but results in a linear increase of inference cost. We evaluate the performance of Spiral under different NFE settings in $\{32, 64, 128, 256, 512, 1024\}$ on SemanticKITTI [3]. The results shown in Figure 7 indicate that Spiral's performance improves significantly when NFE $< 256$, while further increases in NFE yield only marginal gains on most metrics. Therefore, we set NFE $= 256$ as the default configuration. On an A6000 GPU, Spiral achieves an average inference speed of $5.7$ seconds per sample.

**Closed-Loop vs. Open-Loop Inference.** Closed-loop inference is a key innovation in Spiral that enhances cross-modality consistency. To quantify the impact of closed-loop inference, we disable the feedback of the predicted semantic map to Spiral as conditional input, maintaining the open-loop inference throughout the whole generative process. The experimental results on SemanticKITTI [3], as listed in the first row of Table 4, show that adopting the open-loop setting leads to a performance drop of Spiral on all metrics, *e.g.*, a drop of **3.34**% and **8.68**% on S-FRD and S-FPD, respectively.

Table 2: **Comparisons with state-of-the-art LiDAR generation models** on the nuScenes [5] dataset. We evaluate methods using the Range View, Cartesian, and BEV representations. Symbols † and ‡ denote the RangeNet++ [40] backbone and the SPVCNN++ [36] backbone, respectively. The parameter size includes both the generative and segmentation models. The **best** and second best scores under each metric are highlighted in **bold** and underline.

| Method | Param (M) | NFE | Range View FRD↓ (×1) | MMD↓ (×10⁻¹) | S-FRD↓ (×1) | S-MMD↓ (×10⁻¹) | Cartesian FPD↓ (×1) | MMD↓ (×10⁻¹) | S-FPD↓ (×1) | S-MMD↓ (×10⁻¹) | BEV JSD↓ (×10⁻²) | MMD↓ (×10⁻³) | S-JSD↓ (×10⁻²) | S-MMD↓ (×10⁻³) |
|---|---|---|---|---|---|---|---|---|---|---|---|---|---|---|
| LiDARGen† [73] | 30+50 | 1160 | 188.80 | 3.03 | 1381.42 | 29.63 | 26.52 | 12.23 | 1395.83 | 92.31 | 3.98 | 0.15 | 37.37 | 15.23 |
| LiDARGen‡ [73] | 30+97 | | | | 2133.04 | 50.20 | | | 2370.18 | 165.17 | | | 54.43 | 90.78 |
| LiDM† [48] | 275+50 | 256 | — | — | — | — | 124.95 | 90.14 | 1472.81 | 129.65 | 4.00 | 0.29 | 27.60 | 11.75 |
| LiDM‡ [48] | 275+97 | | | | — | — | | | 1823.77 | 147.36 | | | 32.42 | 13.61 |
| R2DM† [42] | 31+50 | 256 | 157.52 | 2.96 | 871.25 | 19.14 | 16.86 | 5.61 | 866.33 | 75.36 | 3.12 | **0.05** | 17.14 | 4.94 |
| R2DM‡ [42] | 31+97 | | | | 1108.43 | 25.27 | | | 1198.16 | 100.38 | | | 35.83 | 17.70 |
| SPIRAL (Ours) | 61 | 256 | **153.40** | **2.95** | **444.04** | **5.49** | **7.13** | **1.04** | **278.64** | **12.86** | **3.10** | **0.05** | **9.12** | **0.80** |

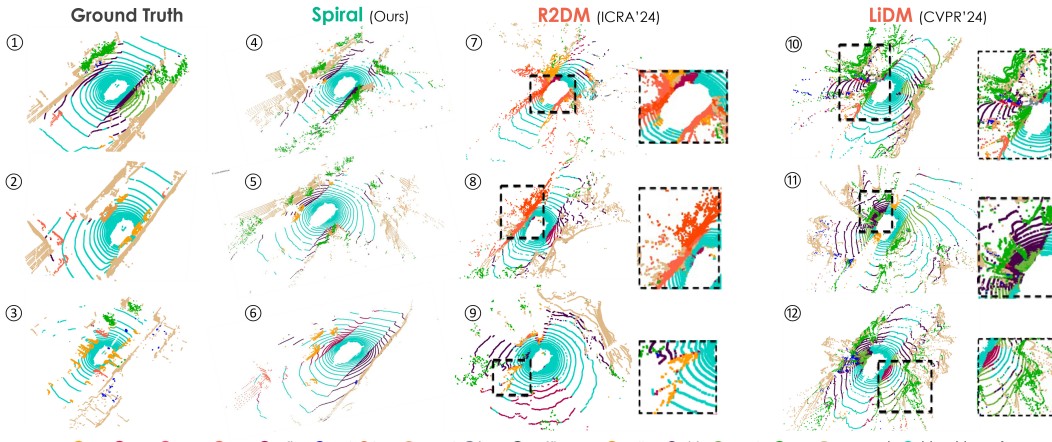

Figure 6: **Visualizations of generated LiDAR scenes** on nuScenes [5]. For two-step methods, we use the labels produced by RangeNet++ [40] due to its superior performance over SPVCNN++ [36]. Artifacts are highlighted with dashed boxes. Examples of semantic artifacts are shown in ⑧, ⑪, and ⑫, while geometric artifacts such as local distortion and large noise are illustrated in ⑦, ⑨, and ⑩.

**Impact of the Confidence Threshold in Closed-loop Inference.** In the closed-loop mode, Spiral adopts a confidence-based filtering strategy to exclude unreliable semantic maps that frequently occur during the early stages of the denoising process. To quantify the effect of the confidence threshold $\delta$, we evaluate the performance of Spiral under different $\delta$ settings in {0.3, 0.5, 0.6, 0.7, 0.8, 0.9}. The results listed in Table 4 indicate that Spiral performs well when $\delta \in$ {0.7, 0.8, 0.9} and achieves slightly best performance at $\delta = 0.8$. Therefore, we set $\delta = 0.8$ as the default configuration of Spiral. However, the performance of Spiral starts to deteriorate when $\delta < 0.6$. With $\delta = 0.3$, the performance of the closed-loop inference even falls behind that of the open-loop inference.

Table 3: **Generative Data Augmentation (GDA) for segmentation training.** We assess GDA using synthetic samples from R2DM [42] and Spiral, under different ratios (1%, 10%, 20%) of real labeled data from SemanticKITTI [3], as well as fog and wet-ground scenes from Robo3D [24]. Symbol † denotes using RangeNet++ as the segmentation model. Results are reported in mIoU (%).

| Setting | 1% | | | 10% | | | 20% | | |
|---|---|---|---|---|---|---|---|---|---|
| GDA | w/o | R2DM† [42] | SPIRAL | w/o | R2DM† [42] | SPIRAL | w/o | R2DM† [42] | SPIRAL |
| SemanticKITTI | 37.76 | 44.16 | **47.41** | 59.07 | 60.62 | **61.14** | 61.16 | 61.21 | **62.35** |
| Robo3D (fog) | 32.07 | 36.82 | **44.06** | 53.93 | 54.20 | **58.61** | 55.89 | 56.07 | **61.24** |
| Robo3D (wet-ground) | 34.26 | 37.96 | **44.19** | 54.70 | 55.92 | **59.31** | 56.81 | 57.53 | **62.02** |

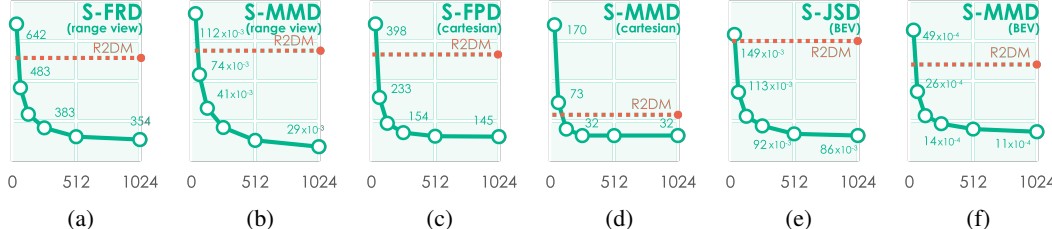

Figure 7: **Impact of the number of function evaluations (NFE).** Increasing the NFE (32, 64, 128, 256, 512, and 1024) improves the performance across all metrics. With fewer sampling steps, Spiral outperforms R2DM [42] using its default setting of NFE = 256, indicated by the dashed line. **Note:** The x-axis denotes NFE, and the y-axis denotes the evaluation metric (lower is better).

Table 4: **Impact of the closed-loop inference and confidence threshold.** The **best** and second best scores under each metric are highlighted in **bold** and underline. The highlighted row indicates the default configuration of Spiral.

| Close-loop | Confidence Threshold $\delta$ | Range View FRD↓ ($\times 1$) | MMD↓ ($\times 10^2$) | S-FRD↓ ($\times 1$) | S-MMD↓ ($\times 10^2$) | Cartesian FPD↓ ($\times 1$) | MMD↓ ($\times 10$) | S-FPD↓ ($\times 1$) | S-MMD↓ ($\times 10$) | BEV JSD↓ ($\times 10^2$) | MMD↓ ($\times 10^3$) | S-JSD↓ ($\times 10^2$) | S-MMD↓ ($\times 10^3$) |
|---|---|---|---|---|---|---|---|---|---|---|---|---|---|
| x | - | 173.93 | 4.98 | 395.64 | 4.35 | 10.61 | 1.82 | 166.95 | 4.41 | 3.89 | 0.16 | 9.29 | 1.44 |
| y | 0.3 | 190.03 | 5.56 | 444.31 | 6.06 | 47.67 | 17.97 | 265.79 | 17.65 | 4.68 | 0.19 | 11.31 | 2.51 |
| y | 0.5 | 174.04 | 5.15 | 396.71 | 4.67 | 22.31 | 6.81 | 187.90 | 8.51 | 3.96 | 0.16 | 9.55 | 1.71 |
| y | 0.6 | 173.25 | 5.02 | 392.36 | 4.24 | 11.14 | 2.59 | 174.25 | 5.12 | 3.87 | **0.15** | 9.40 | 1.59 |
| y | 0.7 | 170.93 | 4.87 | 385.05 | 4.17 | 8.32 | 1.60 | 161.47 | 3.77 | 3.89 | 0.15 | 9.22 | 1.50 |
| **y** | **0.8** | **170.18** | **4.81** | **382.87** | **4.10** | **8.06** | **1.10** | **153.61** | **3.20** | **3.76** | 0.15 | **9.16** | **1.41** |
| y | 0.9 | 170.72 | 5.00 | 384.42 | 4.16 | 8.36 | 1.22 | 155.28 | 3.27 | 3.89 | 0.16 | 9.18 | 1.42 |

**Open-/Closed-Loop Mode Switching Point.** We performed a statistical analysis on 1,000 samples each from SemanticKITTI and nuScenes. With the default 256 denoising steps, closed-loop inference is activated (i.e., once $\geq 80\%$ of semantic predictions exceed the confidence threshold) at an average step of $180 \pm 36$ on SemanticKITTI and $189 \pm 39$ on nuScenes. This indicates that switching typically occurs during the last $\sim 30\%$ of the generation process, when semantic predictions have stabilized, thereby avoiding early-stage noise and ensuring reliable semantic–geometric alignment.

**Inference Efficiency.** We report the average sampling time per sample for LiDARGen [73], LiDM [48], R2DM [42], and **SPIRAL** on an A6000 GPU in Table 5. Additionally, the inference times per sample for RangeNet++ [40] and SPVCNN++ [36] are 0.08s and 0.05s respectively on the same hardware. Unlike the two-step methods, Spiral does not require a segmentation model to generate semantic labels. Spiral demonstrates superior inference efficiency compared to LiDM and LiDARGen. Although it is approximately 2.05 s slower than R2DM combined with SPVCNN++ when using the same number of generation steps, Spiral remains 0.75 s faster with 128 steps and achieves higher generative quality.

Table 5: **Average sampling time per sample** of LiDARGen [73], LiDM [48], R2DM [42], and Spiral on an Nvidia A6000 GPU.

| Method | NFE | Average Time (s) |
|---|---|---|
| LiDARGen [73] | 1160 | 72.0 |
| LiDM [48] | 256 | 7.2 |
| R2DM [42] | 256 | 3.6 |
| **SPIRAL** | 256 | 5.7 |
| **SPIRAL** | 128 | 2.9 |

## 5 Conclusion

We present **SPIRAL**, the first semantic-aware range-view LiDAR diffusion model that jointly generates depth, reflectance, and semantic labels in a unified framework. We further introduce novel semantic-aware evaluation metrics, enabling a holistic assessment of generative quality. Through extensive evaluations on SemanticKITTI and nuScenes, Spiral achieves state-of-the-art performance across multiple geometric and semantic-aware metrics with minimal model size. Additionally, Spiral demonstrates strong utility in downstream segmentation via generative data augmentation, reducing the reliance on manual annotations. We believe Spiral offers a new perspective on multi-modal LiDAR scene generation and opens up promising directions for scalable, label-efficient 3D perception.

## Acknowledgments

The authors would like to sincerely thank the Program Chairs, Area Chairs, and Reviewers for the time and effort devoted during the review process.

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

## Appendix

## 6  Evaluation Metrics

In the main paper, we evaluate the generative quality of Spiral for the LiDAR scene $x$ with semantic map $y$ from three perspectives: range-view images, Cartesian point clouds, and BEV projections. Here, we elaborate on the details of evaluations on each representation.

### 6.1  Evaluation on Range View and Cartesian Point Clouds

For range-view image-based and point cloud-based evaluations, we extract a unified semantic-aware feature $f^s$ by concatenating the geometric feature $\mathcal{E}(x)$, extracted by RangeNet++ [40] or

PointNet [47], with the semantic feature $\mathcal{G}(y)$ from the conditional module in LiDM [48]:

$$f^s \leftarrow \mathcal{E}(x) \oplus \mathcal{G}(y). \tag{10}$$

The features from real and generated sets, $\{f^s\}_r$ and $\{f^s\}_g$, are used to compute S-FRD, S-FPD, and S-MMD. The calculation formula for S-FRD is as follows:

$$\text{S-FRD} = \|\mu_r - \mu_g\|_2^2 + \text{Tr}\left(\Sigma_r + \Sigma_g - 2\left(\Sigma_r\Sigma_g\right)^{\frac{1}{2}}\right), \tag{11}$$

where $\mu_r$ and $\mu_s$ are the mean of $\{f^s\}_r$ and $\{f^s\}_g$, $\Sigma_r$ and $\Sigma_g$ are the covariance matrices, and $\text{Tr}(\cdot)$ is the matrix trace. The calculation of S-FPD follows the same formula. For S-MMD, it can be measured through the kernel trick:

$$\text{S-MMD} = \frac{1}{N^2}\sum_i^N\sum_{i'}^N k(f_i^s, f_i^{s'}) - \frac{2}{NM}\sum_i^N\sum_j^M k(f_i^s, f_j^s) + \frac{1}{M^2}\sum_j^M\sum_{j'}^M k(f_j^s, f_j^{s'}), \tag{12}$$

where $N$ and $M$ are the number of samples in the real and generated sets, respectively. Following [42], we use 3rd-order polynomial kernel function:

$$k(f^s, f^{s'}) = (\frac{(f^s)^T \cdot f^{s'}}{d_f} + 1)^3, \tag{13}$$

where $d_f$ is the dimension of $f^s$.

## 6.2   Evaluation on Bird's Eye View

For the BEV-based evaluation, we first compute the semantic-aware histogram for the real and generated sets, $\{h^s\}_r$ and $\{h^s\}_g$, and then compute the BEV-based S-JSD and S-MMD. The calculation of S-JSD is as follows:

$$\text{JSD}(P||Q) = \frac{1}{2}D_{KL}(P||M) + \frac{1}{2}D_{KL}(Q||M), \tag{14}$$

where $P$ and $Q$ are the approximation of Gausian distribution on $\{h^s\}_r$ and $\{h^s\}_g$, and $M$ is the mean of them: $M = \frac{1}{2}(P + Q)$.

# 7   Data Preprocessing

In this section, we provide more details regarding data preprocessing. For both SemanticKITTI [3] and nuScenes [5], we first project the raw point cloud to range-view images including the depth, reflectance remission, and semantic channels. Then we rescale the depth and reflectance remission channels and encode the semantic maps to RGB images.

## 7.1   Rescaling of the Depth and Reflectance Remission Channels

For the depth channel, we first convert the range-view depth images $x^d$ to log-scale representation $x_{\log}^d$ as follows:

$$x_{\log}^d = \frac{\log(x^d + 1)}{\log(x_{\max}^d + 1)}, \tag{15}$$

where $x_{\max}^d$ is the maximum depth value. Then we linearly rescale the reflectance remission and log-scale depth images to $[-1, 1]$.

## 7.2   Semantic Encoding

For the semantic channels, we use the official color scheme to convert the one-hot semantic map $y \in \mathbb{R}^{H \times W \times C}$ into 2D RGB images, where $C$ is the number of categories. Notably, Spiral is trained in both conditional and unconditional modes, where RGB images in the unconditional mode are replaced with zero padding. To distinguish between zero-padded images from the unconditional mode and the black regions in semantic images from the conditional mode, we add an additional channel to indicate whether a semantic map is provided.

# 8 Model Architecture Details

## 8.1 Spatial Feature Encoding

We adopt a 4-layer Efficient U-Net [50] as the backbone of Spiral, with intermediate feature dimensions of 128, 256, 384, and 640, respectively. Each layer consists of three residual blocks and downsamples the spatial resolution by a factor of two along both the row and column dimensions. The prediction heads for generative residual and semantic predictions are composed of a 2D convolutional layer followed by a two-layer MLP.

## 8.2 Temporal Feature Encoding

For temporal and coordinate encoding, we follow the same strategy as R2DM [42]. For coordinate encoding, we map the per-pixel azimuth and elevation angles to 32-dimensional Fourier features [53], which are then concatenated with the input data. Specifically, the diffusion timestep is encoded to a 256-dimensional sinusoidal positional embedding, which is then integrated by the adaptive group normalization (AdaGN) [9] modules in each layer.

# 9 Discussions of SoTA Segmentation Models in Two-Step LiDAR Generation

In the two-step methods, we report results using both RangeNet++ [40] and SPVCNN++ [36] as segmentation backbones in the main paper. Although SPVCNN++ outperforms RangeNet++ on real-world datasets, it performs worse on the generated LiDAR scenes of LiDARGen [73] and LiDM [48]. In this section, we further present the performance of RangeNet++, SPVCNN++, and a state-of-the-art segmentation model, RangeViT [1], on the generated LiDAR scenes of LiDARGen. As we observe that larger-scale jittering can improve SPVCNN++'s robustness, we train all models with both the default jittering and an increased jittering scale.

## 9.1 Experimental Setup

We use the official implementation of all models. In the default setting, RangeNet++ and RangeViT are trained without jittering augmentation, while SPVCNN++ is trained with the Gaussian noise jittering, where $\sigma = 0.1$. In the setting of larger-scale jittering, we increase the $\sigma$ of jittering to 0.3 across all models.

## 9.2 Observations and Analyses

Detailed results of RangeNet++, SPVCNN++, and RangeViT under different jittering scales are presented in Table 6. Although fine-tuning SPVCNN++ and RangeViT with stronger jittering ($\sigma = 0.3$) improves its performance, they still lag behind RangeNet++, while RangeNet++ presents the best robustness on the generated samples. These experimental results indicate that segmentation models achieving high performance on "clean" real-world scenes do not necessarily perform well on generated scenes. This highlights the limitation of using predictions from state-of-the-art segmentation models as pseudo "ground truth" for evaluating generated scenes. Instead, measuring the distributional similarity of semantic-aware features between real and generated scenes provides a more reliable assessment of the quality of predicted semantic maps, as proposed in this work.

# 10 Additional Qualitative Results

In this section, we present additional visualizations of the generated reflectance remission images, along with more examples of generated LiDAR scenes and the corresponding semantic maps.

## 10.1 Visualizations of Reflectance Remission Images

We visualize the generated reflectance remission images on SemanticKITTI [3] in Figure 8. Since LiDM does not produce reflectance remission images, we only present results from Spiral, LiDARGen [73], and R2DM [42]. Among these, Spiral demonstrates strong cross-modal consistency across depth, reflectance intensity, and semantic labels. In contrast, scenes generated by LiDARGen

Table 6: **Performance of RangeNet++ [40], SPVCNN++ [36], and RangeViT [1] trained with different jittering scales** on the SemanticKITTI [3] dataset.

| Generative Model | Segmentation Backbone | Jittering Scale | Range View | | Cartesian | | BEV | |
|---|---|---|---|---|---|---|---|---|
| | | | S-FRD↓ ($\times 1$) | S-MMD↓ ($\times 10^{-2}$) | S-FPD↓ ($\times 1$) | S-MMD↓ ($\times 10^{-1}$) | S-JSD↓ ($\times 10^{-2}$) | S-MMD↓ ($\times 10^{-3}$) |
| LiDARGen [73] | RangeNet++ [40] | default | 1216.61 | 35.65 | 710.79 | 52.71 | 28.65 | 10.96 |
| | | large | 1202.45 | 34.97 | 702.38 | 50.03 | 28.49 | 10.85 |
| | SPVCNN++ [36] | default | 1978.13 | 70.33 | 1826.54 | 210.67 | 56.40 | 68.97 |
| | | large | 1708.05 | 61.42 | 1366.17 | 103.12 | 54.84 | 68.16 |
| | RangeViT [1] | default | 2034.15 | 72.09 | 1726.54 | 195.44 | 55.20 | 67.47 |
| | | large | 1891.42 | 70.17 | 1625.27 | 187.02 | 54.13 | 65.09 |

and R2DM often exhibit artifacts and inconsistencies across modalities, as highlighted by the red bounding boxes.

## 10.2 Visualizations of Generated LiDAR Scenes

We demonstrate more generated LiDAR scenes in SemanticKITTI [3] and nuScenes [5] in Figure 9 and Figure 10, respectively. The geometric and semantic artifacts of the generated LiDAR scenes are highlighted by dashed boxes.

# 11 Generative Data Augmentation

## 11.1 Full-supervision Setup

In Table 7, we report the experimental results of using the samples generated by a two-step method (RangeNet++ [40] & R2DM [42]) and Spiral for generative data augmentation in the segmentation training of SPVCNN++ [36] on SemanticKITTI [3] dataset. We train SPVCNN++ using full-supervision method. Besides different ratios (1%, 10%, 20%) of available real labeled data, we extend the real data subsets with the generated samples (10k). The results show that the generated samples from Spiral consistently improve the performance of SPVCNN++ and outperform those from R2DM.

## 11.2 Semi-supervision Setup

Two switches in Spiral, $\mathcal{A}$ and $\mathcal{B}$, provide high flexibility to alternate between the unconditional and conditional modes. In fully-supervised training, where all input samples have semantic labels, $\mathcal{A}$ and $\mathcal{B}$ operate in an exclusive-or (X-OR) manner: (1) in the unconditional mode, $\mathcal{A}$ is off and $\mathcal{B}$ is on; (2) in the conditional mode, $\mathcal{A}$ is on and $\mathcal{B}$ is off. When both switches are turned off ($\mathcal{A} = 0$ and $\mathcal{B} = 0$), Spiral degenerates into a normal unconditional LiDAR generative model, *i.e.*, **non-labeled mode**. This design enables Spiral to support semi-supervised training. Specifically, when only a small fraction of training samples are labeled (*e.g.*, 10%), Spiral is trained under labeled data using the conditional or unconditional modes described in the main paper and is trained with the remaining unlabeled samples under the non-labeled mode.

We train Spiral on a training set where only 10% of the samples are labeled, with the remaining samples unlabeled. For the two-step baseline methods, the generative model is trained on the full training set, while the segmentation models are trained solely on the labeled 10% subset. The results presented in Table 8 show that the generative performance of Spiral consistently outperforms other baseline two-step methods in this case.

Additionally, we leverage the samples generated by Spiral, trained under the semi-supervised setting, to augment the labeled training set within the state-of-the-art semi-supervised LiDAR segmentation framework, LaserMix [25]. We adopt MinkUNet [8] and FIDNet [66] as the segmentation backbones. The experimental results in Table 9 demonstrate that incorporating Spiral's generated samples further improves the performance of both MinkUNet and FIDNet within the LaserMix framework.

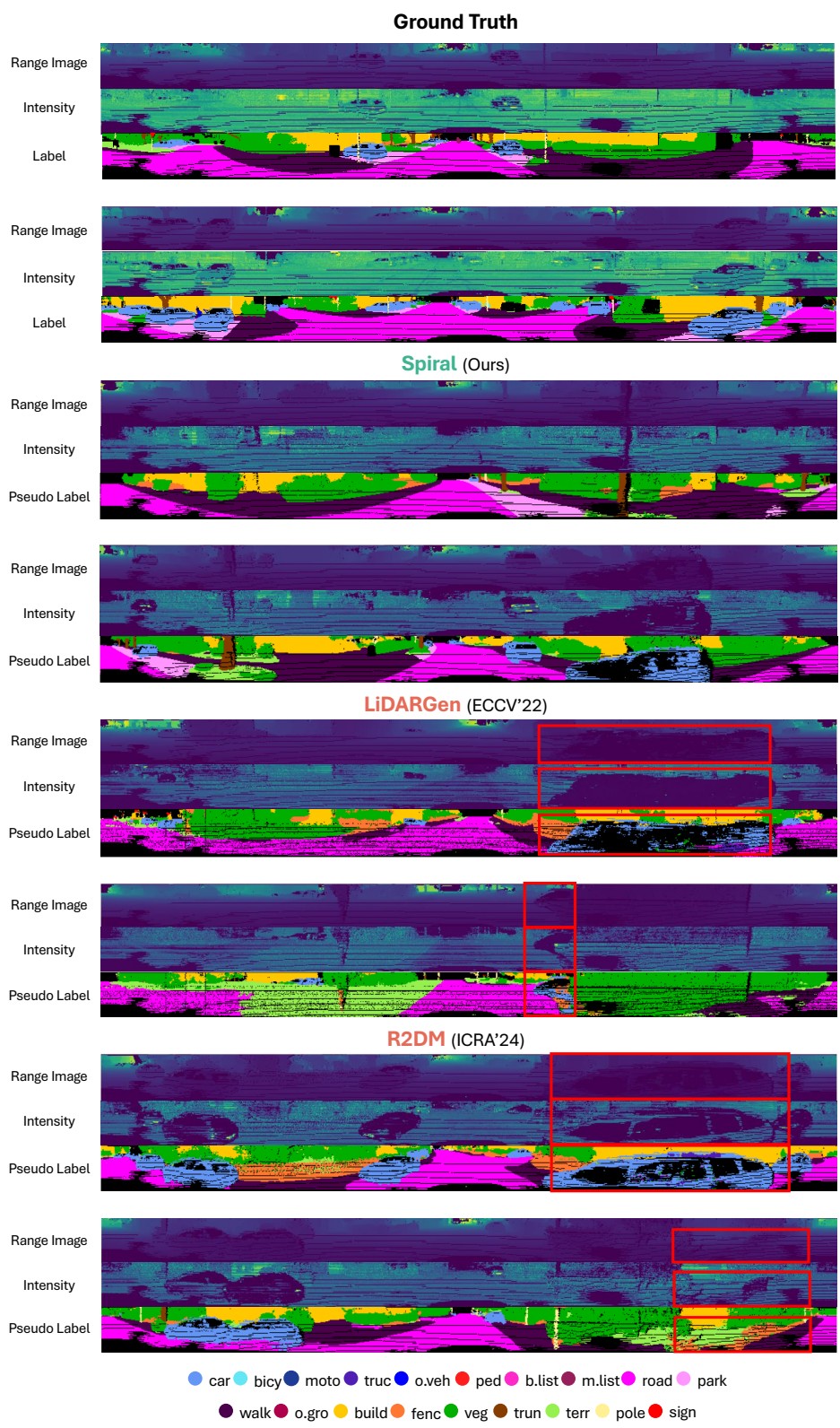

Figure 8: **Visualizations of generated depth, reflectance remission, and semantic labels in range-view perspective** on SemanticKITTI [3]. Generation artifacts and cross-modal inconsistencies are highlighted by the bounding boxes.

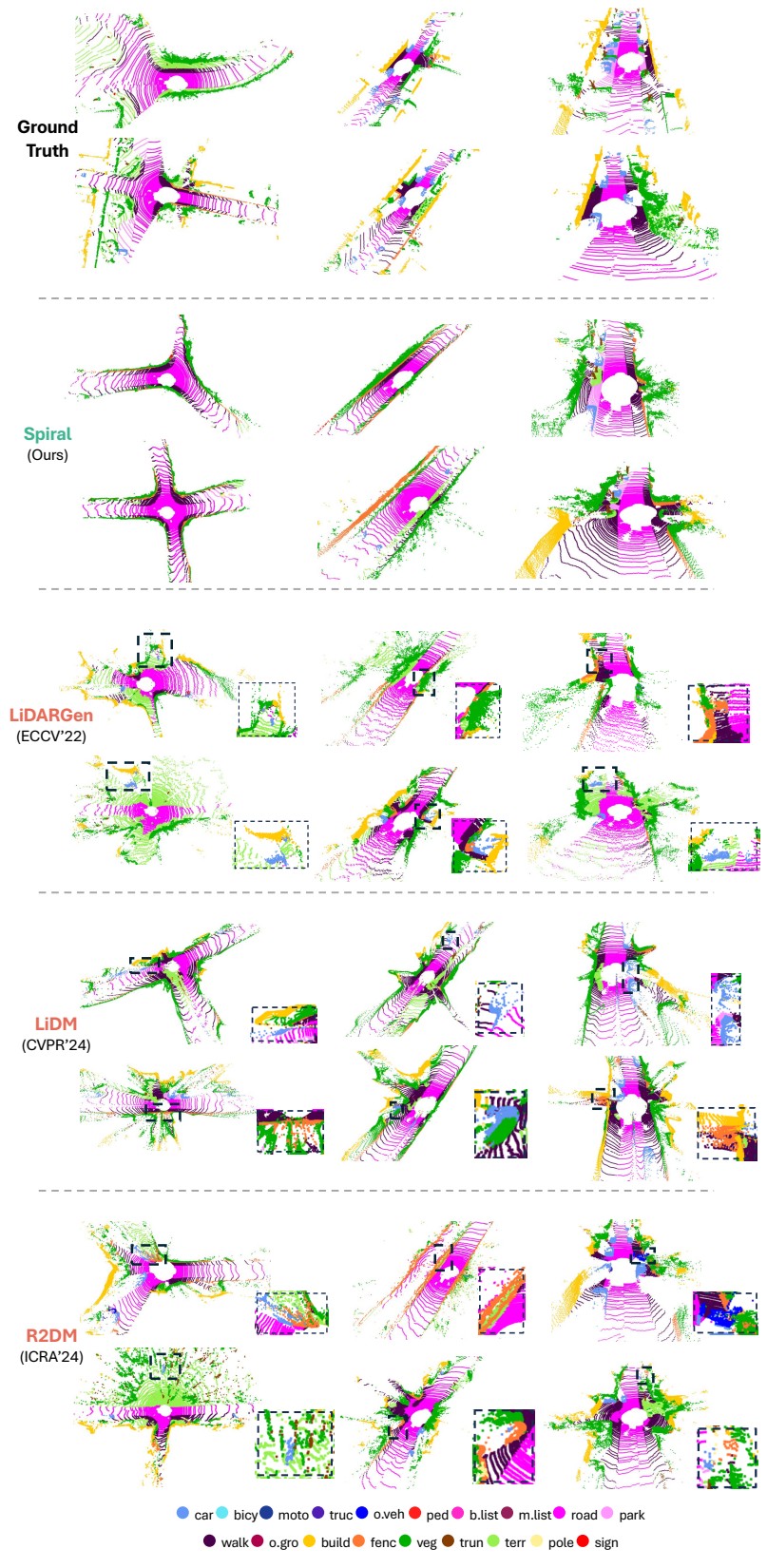

Figure 9: **Visualizations of generated LiDAR scenes** on SemanticKITTI [3].

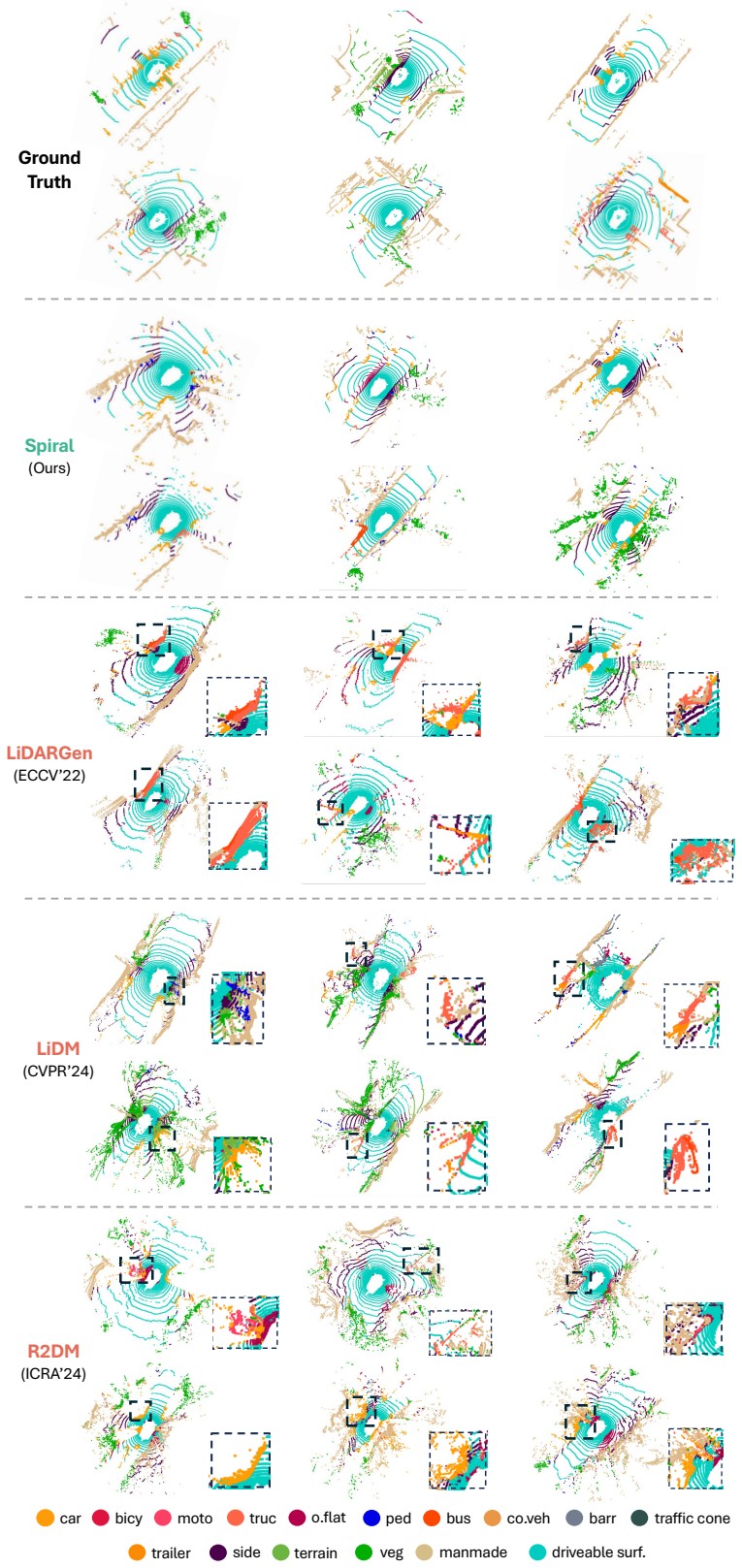

Figure 10: **Visualizations of generated LiDAR scenes** on nuScenes [5].

Table 7: **Generative Data Augmentation (GDA) for segmentation training.** We evaluate the effectiveness of GDA using synthetic samples generated by R2DM [42] & RangeNet++ [40] and Spiral, under different ratios (1%, 10%, 20%) of available real labeled data. Symbol † indicates that RangeNet++ [40] is used as the segmentation backbone.

| Setting (GDA) | | 1% | | | 10% | | | 20% | | |
|---|---|---|---|---|---|---|---|---|---|---|
| | | None | R2DM$^\dagger$ | SPIRAL | None | R2DM$^\dagger$ | SPIRAL | None | R2DM$^\dagger$ | SPIRAL |
| **mIoU** | | 37.76 | 44.16 | 47.41 | 59.07 | 60.62 | 61.14 | 61.16 | 61.21 | 62.35 |
| **per-class mIoU** | car | 89.82 | 87.65 | 92.87 | 94.52 | 94.53 | 95.13 | 95.78 | 94.52 | 96.03 |
| | bicycle | 0.00 | 0.00 | 12.67 | 25.04 | 43.23 | 31.57 | 25.92 | 42.62 | 34.07 |
| | motorcycle | 13.98 | 23.36 | 25.68 | 62.78 | 55.99 | 55.47 | 66.22 | 58.98 | 58.50 |
| | truck | 24.15 | 7.93 | 47.92 | 52.89 | 49.40 | 60.19 | 50.53 | 48.44 | 57.82 |
| | other-vehicle | 15.50 | 29.60 | 33.71 | 42.06 | 52.30 | 53.58 | 55.85 | 51.19 | 55.44 |
| | person | 25.82 | 28.56 | 26.18 | 68.32 | 60.25 | 61.87 | 69.48 | 65.98 | 73.85 |
| | bicyclist | 0.00 | 53.54 | 0.00 | 82.72 | 77.55 | 86.07 | 88.02 | 83.32 | 85.20 |
| | motorcyclist | 1.97 | 0.00 | 0.20 | 0.00 | 0.26 | 0.00 | 0.00 | 1.52 | 0.00 |
| | road | 78.37 | 86.14 | 92.79 | 91.59 | 92.55 | 93.23 | 92.95 | 92.64 | 93.42 |
| | parking | 15.41 | 18.99 | 51.05 | 38.99 | 43.49 | 47.73 | 43.61 | 43.03 | 47.94 |
| | sidewalk | 60.78 | 70.56 | 79.61 | 77.87 | 79.90 | 80.62 | 79.96 | 79.72 | 80.52 |
| | other-ground | 0.01 | 0.25 | 0.67 | 4.21 | 2.09 | 1.30 | 2.11 | 3.31 | 1.40 |
| | building | 79.20 | 86.93 | 88.38 | 88.58 | 90.40 | 90.52 | 89.87 | 90.13 | 90.03 |
| | fence | 32.62 | 52.14 | 53.14 | 56.13 | 62.22 | 61.82 | 59.80 | 60.77 | 60.45 |
| | vegetation | 83.62 | 86.28 | 88.07 | 87.97 | 89.49 | 89.82 | 88.35 | 89.21 | 88.69 |
| | trunk | 52.78 | 46.19 | 61.62 | 66.11 | 66.58 | 68.27 | 66.29 | 66.69 | 68.51 |
| | terrain | 69.38 | 72.84 | 77.67 | 74.97 | 78.36 | 78.28 | 75.63 | 77.74 | 75.34 |
| | pole | 54.32 | 44.96 | 54.26 | 63.03 | 61.80 | 62.81 | 64.27 | 62.12 | 63.85 |
| | traffic-sign | 19.77 | 43.12 | 14.21 | 44.61 | 51.44 | 43.44 | 47.39 | 51.01 | 53.66 |

Table 8: **Semi-supervised training of Spiral and the baseline two-step methods** on the SemanticKITTI [3] dataset. We evaluate methods using the Range View, Cartesian, and BEV representations. Symbols † denotes the RangeNet++ [40] trained with 10% labeled samples. The parameter size includes both the generative and segmentation models.

| Method | Param (M) | NFE | Range View | | Cartesian | | BEV | |
|---|---|---|---|---|---|---|---|---|
| | | | S-FRD↓ | S-MMD↓ | S-FPD↓ | S-MMD↓ | S-JSD↓ | S-MMD↓ |
| | | | $\times 1$ | $\times 10^{-2}$ | $\times 1$ | $\times 10^{-1}$ | $\times 10^{-2}$ | $\times 10^{-3}$ |
| LiDARGen$^\dagger$ [73] | 30+50 | 1160 | 1285.55 | 47.56 | 750.68 | 58.26 | 34.49 | 12.66 |
| LiDM$^\dagger$ [48] | 275+50 | 256 | – | – | 486.22 | 29.78 | 16.91 | 6.33 |
| R2DM$^\dagger$ [42] | 31+50 | 256 | 528.82 | 7.82 | 281.96 | 8.95 | 16.13 | 2.99 |
| SPIRAL | **61** | 256 | **508.86** | **5.77** | **211.97** | **4.79** | **11.47** | **2.65** |

# 12 Broader Impact & Limitations

In this section, we elaborate on the broader impact, societal influence, and potential limitations of the proposed approach.

## 12.1 Broader Impact

This work presents several significant implications for both academic research and practical applications in autonomous driving and computer vision. SPIRAL advances the field of LiDAR scene generation by introducing a unified framework for joint depth, reflectance, and semantic label generation. This breakthrough could substantially impact:

**1. Research Community**

Table 9: **Generative Data Augmentation (GDA) for semi-supervised training in the Laser-Mix [25] framework.** We evaluate the effectiveness of GDA using synthetic samples generated by Spiral in the semi-supervised training framework LaserMix.

| Backbone | MinkUNet [8] | | | FIDNet [66] | | |
|---|---|---|---|---|---|---|
| Method | sup. only | LaserMix [25] | **LaserMix +SPIRAL** | sup. only | LaserMix [25] | **LaserMix +SPIRAL** |
| mIoU | 64.0 | 66.6 | **67.2** | 52.2 | 60.1 | **60.3** |

- Establishes new benchmarks for semantic-aware LiDAR generation
- Provides a novel framework for multi-modal scene understanding
- Opens new research directions in 3D scene generation and understanding

**2. Industry Applications**

- Reduces the dependency on expensive and time-consuming manual data annotation
- Enables more efficient development of autonomous driving systems
- Provides a cost-effective solution for synthetic data generation

## 12.2  Societal Influence

The development of SPIRAL could lead to several positive societal outcomes:

**1. Transportation Safety**

- Enhanced autonomous vehicle perception capabilities through better training data
- Potential reduction in traffic accidents through improved scene understanding

**2. Economic Impact**

- Reduced development costs for data collection
- Accelerated deployment of self-driving vehicles

## 12.3  Potential Limitations

While SPIRAL demonstrates promising results, several limitations and challenges should be acknowledged:

**1. Technical Constraints:** The quality of generated scenes may still have room for improvement.

**2. Methodological Limitations:** Generated data may not fully capture all real-world complexities.

**3. Implementation Challenges:** Real-time performance considerations for practical applications.

This work represents a significant step forward in semantic-aware LiDAR scene generation while acknowledging the need for continued research to address these limitations and challenges. Future work could focus on enhancing the model's capabilities in handling complex scenarios and improving computational efficiency while maintaining generation quality.

# 13   Public Resource Used

In this section, we acknowledge the use of the public resources, during the course of this work:

## 13.1   Public Datasets Used

- nuScenes[2] . . . . . . . . . . . . . . . . . . . . . . . . . . . . . . . . . . . . . . . . . . . . . . . . . . . . . . . . . . CC BY-NC-SA 4.0

---

[2]https://www.nuscenes.org/nuscenes.

- nuScenes-DevKit[3] .......................................... Apache License 2.0
- SemanticKITTI[4] .......................................... CC BY-NC-SA 4.0
- SemanticKITTI-API[5] ............................................ MIT License
- Robo3D[6] ................................................. CC BY-NC-SA 4.0

## 13.2 Public Implementations Used

- MMDetection[7] ........................................... Apache License 2.0
- MMDetection3D[8] ......................................... Apache License 2.0
- RangeNet++[9] ................................................... MIT License
- OpenPCSeg[10] ............................................ Apache License 2.0
- RangeViT[11] ............................................. Apache License 2.0
- LiDARGen[12] .................................................... MIT License
- LiDM[13] ....................................................... MIT License
- R2DM[14] ....................................................... MIT License

---

[3] https://github.com/nutonomy/nuscenes-devkit.
[4] http://semantic-kitti.org.
[5] https://github.com/PRBonn/semantic-kitti-api.
[6] https://github.com/worldbench/robo3d.
[7] https://github.com/open-mmlab/mmdetection.
[8] https://github.com/open-mmlab/mmdetection3d.
[9] https://github.com/PRBonn/lidar-bonnetal.
[10] https://github.com/PJLab-ADG/OpenPCSeg.
[11] https://github.com/valeoai/rangevit.
[12] https://github.com/vzyrianov/lidargen.
[13] https://github.com/hancyran/LiDAR-Diffusion.
[14] https://github.com/kazuto1011/r2dm.

