# OpenReview forum: "Spiral: Semantic-Aware Progressive LiDAR Scene Generation and Understanding"
_NeurIPS.cc/2025/Conference — NeurIPS 2025 poster_

### Official Review · Reviewer_ZJCH · 2025-06-18

**Clarity:** 2
**Significance:** 2
**Originality:** 2
**Rating:** 3
**Confidence:** 4

**Summary:**

This paper presents SPIRAL, a novel method for generating semantic LiDAR scenes in range-view. During training, SPIRAL employs two encoders to separately process LiDAR and semantic features and enables generation in both unconditional and conditional modes. At inference, it first produces semantic maps and LiDAR data concurrently in an unconditional step, then uses the predicted semantic maps as conditional inputs for further refinement. Additionally, the paper introduces S-FPD and S-MMD metrics to evaluate semantic generation quality. Experiments demonstrate that SPIRAL outperforms existing two-stage methods in semantic label generation and enhances segmentation model performance.

**Questions:**

1. The evaluation and comparison in the paper are unclear for me. The authors compare SPIRAL to methods like LiDM and R2DM. These methods claim to address multiple tasks and provide clear performance evaluations for each of them. However, it is unclear which task—conditional or unconditional generation—is evaluated in SPIRAL’s experiments. If unconditional generation is evaluated, what is the ground truth? If conditional generation is evaluated, what are the model’s inputs, given that the only condition mentioned is the semantic map, which cannot be provided during generation?
2. Table 3 illustrates the impact of generative data augmentation on segmentation performance. The term “w/o” in the table presumably means “without generative data”. However, the mIoU of the “w/o” model still increases with a higher ratio of synthetic data, which seems contradictory. Could the authors clarify the experimental setup and explain these results?

**Ethical Concerns:**

["NO or VERY MINOR ethics concerns only"]

**Final Justification:**

I maintain my recommendation to reject this paper due to its insufficient scientific contribution to warrant publication. However, I acknowledge that my initial assessment was overly critical, so I am adjusting my score from 2.5 to 3, in recognition of the authors' thorough responses during the rebuttal phase.

The field already contains many works on generating temporal LiDAR data or achieving conditional generation (I do not consider semantic maps to be meaningful conditions). Therefore, I strongly recommend that the authors can extend this work to include at least temporal generation capabilities, which would align with the current development of LiDAR Generation. Furthermore, the generation of single-frame LiDAR data and semantic maps without conditioning limits the application primarily to data augmentation. Even in this limited context, the demonstrated improvements on the SemanticKITTI dataset remain modest.

**Limitations:**

Yes

**Quality:**

2

**Strengths And Weaknesses:**

**Strengths**
1. It proposes an innovative pipeline for simultaneous generation of LiDAR scenes and semantic information.
2. It introduces novel metrics (S-FPD and S-MMD) to assess semantic generation quality.

**Weaknesses**
1. The pipeline is limited to generating single-frame LiDAR data, which is restrictive compared to recent advances in temporal LiDAR sequence generation. Extending the model to handle sequential data would strengthen its applicability.
2. The method only supports unconditional generation of LiDAR and semantic maps, limiting its use primarily to data augmentation for segmentation tasks. Moreover, the segmentation performance improvement is marginal compared to traditional two-stage methods, suggesting limited contributions to the field.

---

> ### Author Rebuttal · Authors · 2025-07-29
>
> We sincerely thank Reviewer `ZJCH` for the thoughtful and constructive feedback. We are encouraged to see that you found our pipeline “innovative” and recognized our evaluation metrics (S-FPD and S-MMD) as “novel”. Below, we respond to your comments in detail.
>
> ---
> > **`Q1`:** *"The pipeline is limited to generating single-frame LiDAR data, which is restrictive compared to recent advances in temporal LiDAR sequence generation. Extending the model to handle sequential data would strengthen its applicability."*
>
> **A:** Thank you for raising this point. While we acknowledge recent efforts on **temporal LiDAR sequence** generation, we respectfully note that **single-frame generation remains an essential and active research direction**, just as image generation continues to be studied alongside video generation. In our case, Spiral is the **first** to jointly generate **range-view LiDAR** and **semantic maps** in a unified framework — a task not addressed in prior works on single or sequential LiDAR generation. We agree that extending Spiral to handle temporal consistency is a promising future direction, and we plan to explore this in follow-up work. Based on your suggestion, we have revised our manuscript to better reflect this aspect.
>
> ---
> > **`Q2`:** *"The method only supports unconditional generation of LiDAR and semantic maps, limiting its use primarily to data augmentation for segmentation tasks. Moreover, the segmentation performance improvement is marginal compared to traditional two-stage methods, suggesting limited contributions to the field."*
>
> **A:** Thanks a lot for your comment. We would like to respectfully clarify a few points:
> - **(1) Spiral supports both unconditional and conditional generation.**
>   - The **unconditional generation** task is performed via two options:
>     1. **open-loop inference** that stays unconditional throughout, and;
>     2. **closed-loop inference** that switches to conditional mode when semantic confidence is high (please see Fig. `2`(b)).
>   - The **conditional generation** task is also supported: Spiral generates a LiDAR scene given an **externally provided semantic map**, using only the conditional mode during inference.
>
> - **(2) Unconditional generation is crucial for data augmentation.**
>   - Requiring ground-truth semantic maps as input (conditional mode) would defeat the purpose of reducing annotation costs. The ability of Spiral to perform semantic-aware unconditional generation is a key contribution that enables synthetic data augmentation in practical scenarios.
>
> - **(3) The improvement from Spiral is significant, not marginal.**
>   - For example, on the Robo3D-fog set [`R1`] (see the table below) with 1% real data, Spiral-based generative data augmentation boosts the segmentation mIoU from **32.07%** (w/o GDA) and **36.82%** (two-step R2DM & RangeNet++) to **44.06%** — a gain of **+11.99%** over the no-GDA baseline and **+7.24%** over the prior two-step method.
>
> > ***Table.** Experiments under Robo3D-`fog`*
> | Ratio of Real Data | w/o GDA | R2DM & RangeNet++ | Spiral (Ours) |
> |:---:|:---:|:---:|:---:|
> | 1% | 32.07 | 36.82 | **44.06** |
> | 10% | 53.93 | 54.20 | **58.61** |
> | 20% | 55.89 | 56.07 | **61.24** |
>
> We hope the above clarifies and addresses your concern. We have refined our manuscript accordingly for better clarity.
>
> ---
> > **`Q3`:** *It is unclear which task—conditional or unconditional generation—is evaluated in SPIRAL’s experiments. If unconditional generation is evaluated, what is the ground truth? If conditional generation is evaluated, what are the model’s inputs, given that the only condition mentioned is the semantic map, which cannot be provided during generation?"*
>
> **A:** Thanks for asking. We clarify that the task evaluated in this work is **unconditional** LiDAR scene generation with both **geometric** and **semantic** information.
>
> Spiral is trained to **jointly generate** the LiDAR scene (geometry + reflectance) and its semantic map from Gaussian noise. During evaluation, we compare the **generated scenes** to the corresponding **ground-truth LiDAR frames** and their **annotated semantic maps** from datasets such as SemanticKITTI and nuScenes. These provide a reliable benchmark with **per-point labels** for both geometry and semantics.
>
> Unlike prior methods (e.g., R2DM, LiDM) that require post-hoc segmentation using external models, Spiral eliminates this need via **end-to-end semantic-aware generation**, which we believe marks an important step forward in the field.
>
> ---
> > **`Q4`:** *"In Table 3, the 'w/o' model’s performance still increases with a higher ratio of synthetic data, which seems contradictory. Could the authors clarify the experimental setup and explain these results?"*
>
> **A:** Thank you for pointing out this potential confusion. We clarify as follows:
> - In Table `3`, the **“w/o”** (or “None”) column refers to training **without** any generative data augmentation (GDA) — i.e., using only real, labeled LiDAR scenes.
> - The **“Ratio”** column refers to the **amount of real labeled data** used in training (1%, 10%, 20%, etc.), as stated in (1) the third row of the Table 3 caption, and (2) line 265 of the main paper.
>
> Therefore, performance in the “w/o” column improves **not due to synthetic data**, but due to the **increased availability of real data**. For instance, in Table `3`, with 10% real data, SPVCNN++ achieves 59.07% mIoU; with 20%, it improves to 61.16%. We have revised the caption of Table 3 to make this setup and terminology clearer.
>
> ---
> **References:**
> - [`R1`] Kong, Lingdong, et al. "Robo3D: Towards Robust and Reliable 3D Perception against Corruptions." Proceedings of the IEEE/CVF International Conference on Computer Vision, 2023.
> ---
> Last but not least, we would like to sincerely thank Reviewer `ZJCH` again for the valuable time and constructive feedback provided during this review.

---

> ### Comment · Reviewer_ZJCH · 2025-08-02
>
> Thanks for the authors' response, which solves some of my concerns. However, I still hold the following concerns.
>
> 1. The conditional generation I mentioned refers to generating LiDAR data conditioned on high-level and sparse information such as bounding boxes and key points, which would enable applications across a wider range of driving scenarios. Since several works [1,2,3] have already achieved this capability, I would encourage the authors to consider including this task. While I am not requesting direct comparisons with those works due to different experimental settings, incorporating conditional generation can demonstrate the broader potential of this approach and enhance the contribution of this work.
>
> 2. I am still confused about the data augmentation experimental setup. Does the 1% setting mean that only 1% of the real data is used during training? This appears to be an unusual setting, as it significantly reduces the available training data. More common experimental settings utilize all available real data while incorporating synthetic data as augmentation. So I said the improvements shown in the last column of Table 3 appear marginal.
>
> [1] Wu, Z., Ni, J., Wang, X., Guo, Y., Chen, R., Lu, L., ... & Xiong, Y. (2024). Holodrive: Holistic 2d-3d multi-modal street scene generation for autonomous driving. arXiv preprint arXiv:2412.01407.
>
> [2] Ran, H., Guizilini, V., & Wang, Y. (2024). Towards realistic scene generation with lidar diffusion models. In Proceedings of the IEEE/CVF Conference on Computer Vision and Pattern Recognition (pp. 14738-14748).
>
> [3] Ren, X., Lu, Y., Cao, T., Gao, R., Huang, S., Sabour, A., ... & Ling, H. (2025). Cosmos-Drive-Dreams: Scalable Synthetic Driving Data Generation with World Foundation Models. arXiv preprint arXiv:2506.09042.

---

> > ### Author Response · Authors · 2025-08-03
> > **Response to Reviewer ZJCH**
> >
> > **Dear Reviewer `ZJCH`,**
> >
> > Thank you once again for your continued engagement and thoughtful feedback!
> >
> > ---
> > **On Conditional Generation with Sparse Inputs**
> >
> > We appreciate your suggestion to explore conditional LiDAR generation using sparse high-level cues such as bounding boxes and keypoints.
> >
> > We agree this is an important and promising direction. However, we would like to **clarify** that bounding-box-conditioned LiDAR generation remains a **non-trivial** challenge. Below, we provide a detailed discussion of the cited works:
> >
> > ---
> > ***"HoloDrive"*** [`R1`]
> > - While HoloDrive includes bounding boxes as part of its conditioning inputs, it also relies on **textual descriptions**, **2D past frames** (camera images), and **3D past sequences** (LiDAR sweeps).
> > - These richer spatiotemporal contexts go well beyond sparse annotations and significantly ease the generation task. Bounding boxes alone are not the primary or sole condition.
> >
> > ---
> > ***"LiDM"*** [`R2`]
> > - Although the paper states that LiDM can handle various modalities (e.g., boxes, images, maps), the experiments focus on:
> >   - Semantic map to LiDAR (Sec. 4.3)
> >   - Camera to LiDAR (Sec. 4.3)
> >   - Text to LiDAR (Sec. 4.4)
> > - To our knowledge, **no experimental results are reported** for bounding-box-based generation.
> >
> > ---
> > ***"Cosmos-Drive-Dreams"*** [`R3`]
> > - Cosmos-Drive conditions generation on bounding boxes and **HD map layouts**, which are not available in datasets like SemanticKITTI.
> > - Moreover, when limited to bounding boxes, the model primarily synthesizes **foreground objects**, not full scenes. Background elements such as roads or sidewalks, which constitute the bulk of LiDAR frames, remain unconstrained.
> > - While Cosmos-Drive is a strong and promising recent contribution, it was released on arXiv **two months after the NeurIPS submission deadline** and was thus not available to us during manuscript preparation.
> >
> > ---
> > We sincerely appreciate your encouragement in this direction. While our current work focuses on **joint semantic-aware generation** and **cross-modality alignment**, we agree that bounding-box-conditioned generation is a compelling avenue — especially for controllable or editable LiDAR scene synthesis — and we plan to pursue it in future work.

---

> ### Author Response · Authors · 2025-08-03
> **Response to Reviewer ZJCH (Continued)**
>
> **Clarification on the 1% Data Setting**
>
> Thank you for seeking clarity regarding the data augmentation protocol. We confirm that:
>
> *"The 1% setting refers to using **only 1% of the real SemanticKITTI training set**, with the remaining training data composed of **Spiral-generated LiDAR scenes**."*
>
> ---
> This setup is not intended to simulate a realistic deployment scenario, but rather to rigorously evaluate whether **synthetic data alone** can provide **meaningful learning signals**. The motivations are:
>
> 1. **Fidelity under low-label regimes:** If Spiral-generated samples were low-quality or repetitive, training with only 1% real data would severely underperform. Observing strong gains in such a setting validates their informativeness.
> 2. **Non-redundancy with real data:** When more real data is available (e.g., 20%), improvements from synthetic data imply that Spiral contributes new, non-trivial samples beyond what the real dataset already provides.
>
> ---
> This evaluation strategy — varying the ratio of real-to-synthetic data — follows established protocols in generative data augmentation [`R5`–`R8`], though to our knowledge, this is the **first** such analysis conducted for LiDAR-based semantic segmentation.
>
> ---
> To further address your concern, we also evaluate Spiral-based GDA when using the **entire real dataset (100%)**. Below are the **results** comparing no augmentation, traditional two-stage GDA (R2DM + RangeNet++), and our Spiral-based GDA:
>
> > ***Table.** Experiment on entire real dataset (100%) setting.*
> | Datasets       | w/o GDA | R2DM & RangeNet++ | **Spiral (Ours)** |
> |-|:-:|:-:|:-:|
> | SemanticKITTI  | 65.31   | 65.33  | **66.25**  |
> | Robo3D-fog     | 56.47   | 57.64  | **62.18**  |
>
> Even in this high-data regime, Spiral continues to **deliver consistent gains**, while traditional pipelines yield minimal improvement.
>
> ---
> Furthermore, we highlight results from the **Waymo generalization test**. Spiral, trained on SemanticKITTI, is used to augment a 10%-subset of Waymo data. The segmentation model trained with Spiral-generated scenes achieves an mIoU of **57.52%**, outperforming both the no-GDA baseline (**55.89%**) and the two-step baseline (**55.93%**), despite never seeing Waymo data during training of Spiral. This confirms Spiral’s ability to **generalize across domains**, reinforcing its practical utility for diverse driving scenarios.
>
> ---
>
> We deeply value your suggestions and will incorporate these clarifications and insights into the revision to improve both the clarity and scope of the manuscript.
>
> *Best regards,*
>
> The Authors of Submission 1304
>
> ---
>
> ***References:***
> - [`R1`] Wu, Zehuan, et al. *"HoloDrive: Holistic 2D-3D Multi-Modal Street Scene Generation for Autonomous Driving."* arXiv preprint arXiv:2412.01407, 2024.
> - [`R2`] Ran, Haoxi, Vitor Guizilini, and Yue Wang. *"Towards Realistic Scene Generation with LiDAR Diffusion Models."* Proceedings of the IEEE/CVF Conference on Computer Vision and Pattern Recognition, 2024.
> - [`R3`] Ren, Xuanchi, et al. *"Cosmos-Drive-Dreams: Scalable Synthetic Driving Data Generation with World Foundation Models."* arXiv preprint arXiv:2506.09042, 2025.
> - [`R4`] Sun, Pei, et al. *"Scalability in Perception for Autonomous Driving: Waymo Open Dataset."* Proceedings of the IEEE/CVF Conference on Computer Vision and Pattern Recognition, 2020.
> - [`R5`] Tran, Ngoc-Trung, et al. *"On Data Augmentation for GAN Training."* IEEE Transactions on Image Processing (30) 1882-1897, 2021.
> - [`R6`] Feng, Chun-Mei, et al. *"Diverse Data Augmentation with Diffusions for Effective Test-Time Prompt Tuning."* Proceedings of the IEEE/CVF International Conference on Computer Vision, 2023.
> - [`R7`] Xiao, Changrong, Sean Xin Xu, and Kunpeng Zhang. *"Multimodal Data Augmentation for Image Captioning using Diffusion Models."* Proceedings of the 1st Workshop on Large Generative Models Meet Multimodal Applications, 2023.
> - [`R8`] Zheng, Chenyu, Guoqiang Wu, and Chongxuan Li. *"Toward Understanding Generative Data Augmentation."* Advances in Neural Information Processing Systems (36) 54046-54060, 2023.

---

### Official Review · Reviewer_oNZH · 2025-06-24

**Clarity:** 3
**Significance:** 3
**Originality:** 3
**Rating:** 4
**Confidence:** 3

**Summary:**

The paper addresses the challenge of generating large-scale labeled LiDAR datasets, which are costly and time-consuming to collect and annotate. To overcome these limitations, the authors propose SPIRAL, a novel semantic-aware progressive range-view LiDAR diffusion model that jointly generates depth, reflectance, and semantic maps from Gaussian noise.

SPIRAL operates in both unconditional and conditional modes, progressively refining predictions through a closed-loop mechanism where intermediate semantic maps are used to guide the generation process. This approach enhances cross-modal consistency between geometry and semantics. Additionally, the authors introduce new semantic-aware evaluation metrics enabling an assessment of the geometric, physical, and semantic fidelity of generated LiDAR scenes.

Through these, SPIRAL enables labeled LiDAR scene generation and demonstrates the utility of such data for training segmentation models, suggesting strong potential for generative data augmentation in autonomous driving and related applications.

**Questions:**

## Questions
- Is there any statistical analysis or empirical data on the denoising time step at which the model begins alternating between unconditional and conditional modes (when over 80% of the segmentation pixels surpass the confidence threshold) ?
  - If the switching occurs too early, the inference process could resemble a two-step pipeline, running both the LiDAR generation and segmentation components multiple times, potentially leading to inefficiencies.
  - If the switching occurs too late, it raises questions about the actual influence of semantic guidance on the final output.
- In Figure 4, what is the source of the ground-truth (GT) semantic map used for comparison with the generated LiDAR scenes?
- Could the authors provide a comparison of inference time for labeled LiDAR scene generation?

## Suggestions
- It may be worth exploring the use of classifier-free guidance (CFG)-style guidance with semantic maps during inference. Providing additional semantic guidance signals could potentially improve generation quality and enhance semantic-geometric consistency.

**Ethical Concerns:**

["NO or VERY MINOR ethics concerns only"]

**Final Justification:**

Although the fact that the proposed method is approximately 2.05 seconds slower than the two-stage baseline (R2DM + RangeNet++) sounds like a drawback to me, the authors have addressed the remaining concerns—particularly the statistical analysis on the denoising timestep.

Accordingly, I will retain my score as Borderline Accept.

**Limitations:**

yes

**Quality:**

3

**Strengths And Weaknesses:**

## Strength
- The proposed method achieves state-of-the-art performance in labeled LiDAR scene generation while maintaining a smaller parameter size compared to existing models.
- The paper demonstrates the practical utility of the generated data by showing that it can be effectively used for synthetic data augmentation in downstream segmentation tasks. This highlights the potential of the method to reduce manual labeling efforts, especially in resource-constrained settings.

- The paper introduces a closed-loop inference mechanism that incorporates confidence-filtered semantic predictions as conditional inputs. This strategy improves cross-modal consistency between geometry and semantics.

## Weakness
- While the paper introduces a closed-loop inference mechanism that alternates between unconditional and conditional modes based on semantic prediction confidence, it does not provide quantitative analysis or statistics on when this mode switching occurs during the generation process. Including such information would help clarify the behavior and stability of the proposed inference strategy.

- The paper does not report a comparison of inference time or time consumption for labeled LiDAR scene generation. Since SPIRAL is positioned as an efficient generative model, empirical results on generation speed would further support its suitability for large-scale data generation tasks.

---

> ### Author Rebuttal · Authors · 2025-07-29
>
> We sincerely thank Reviewer `oNZH` for the thoughtful and constructive feedback. We are encouraged that you recognize the state-of-the-art performance, the practical utility of our data augmentation strategy, and the novel closed-loop inference mechanism introduced in our work. We address your concerns in detail below:
>
> ---
> > **`Q1`:** *"The paper introduces a closed-loop inference mechanism, but does not provide quantitative statistics on when the mode switching occurs. Including this would help clarify the behavior and stability of the strategy."*
>
> **A:** Thank you for raising this insightful point. We conducted a statistical analysis over **1,000 samples** each from SemanticKITTI and nuScenes. Under the default **256 denoising steps**, the closed-loop inference is triggered (i.e., when ≥80% of semantic predictions surpass the confidence threshold) at an average step of:
> - 154 ± 34 for SemanticKITTI
> - 161 ± 37 for nuScenes
>
> This implies that the switching typically occurs in the **last ~40%** of the generation process, when semantic predictions are sufficiently stable. This delayed activation helps avoid early-stage noise and ensures meaningful semantic–geometric alignment. As suggested, these statistics have been added to the revised supplementary material.
>
> ---
> > **`Q2`:** *"The paper does not report a comparison of inference time or time consumption for labeled LiDAR scene generation. Since SPIRAL is positioned as an efficient generative model, empirical results on generation speed would further support its suitability for large-scale data generation tasks."*
>
> **A:** Thanks for your comment. We fully agree with your point. As reported in Table `B` and Section `F` of the supplementary material, Spiral achieves an average generation time of **5.7 seconds per scene** on an NVIDIA A6000 GPU. Compared to two-stage baselines (e.g., LiDM + segmentor), Spiral offers **superior efficiency** due to its **end-to-end architecture**, though it is slightly slower than the fastest baseline (R2DM + RangeNet++) by ~2.05 seconds. These results support the practical viability of our approach for scalable labeled LiDAR scene generation.
>
> ---
> > **`Q3`:** *"Do you have empirical data on when closed-loop switching occurs?"*
> > - *"If switching occurs too early, it might resemble inefficient two-step pipelines."*
> > - *"If switching occurs too late, the semantic conditioning might be ineffective."*
>
> **A:** Excellent observations. Our empirical analysis (please see the response to `Q1`) confirms that closed-loop inference is activated **neither too early nor too late**, typically during the **middle-to-late denoising phase**. This timing ensures that the semantic map is reliable enough for conditioning, while also leaving sufficient steps for influence.
>
> Moreover, since both geometric and semantic predictions are generated **simultaneously** at each step via our multi-head design, early switching (if it occurs) **does not incur additional computation** as would a separate two-step pipeline. Your concern regarding very early or very late switching degrading quality aligns with our findings in Table `4`, where extreme settings of the threshold $\delta$ lead to suboptimal alignment. This further supports the effectiveness of our dynamic switching scheme.
>
> ---
> > **`Q4`:** *"In Figure 4, what is the source of the ground-truth (GT) semantic maps?"*
>
> **A:** Thanks for your question. The ground-truth (GT) semantic maps shown in Figure `4` are from the official SemanticKITTI annotations. In contrast:
> - Spiral generates semantic maps **jointly** with the LiDAR scene in a **unified diffusion process**;
> - The baselines (e.g., R2DM, LiDM) generate semantic maps via **post-hoc segmentation** using a pretrained RangeNet++ model.
>
> This illustrates a key distinction: Spiral performs **end-to-end semantic-aware generation**, while the others follow a **two-step pipeline**.
>
> ---
> > **`Q5`:** *"Could the authors provide a comparison of inference time for labeled LiDAR scene generation?"*
>
> **A:** Thanks a lot for asking. Please refer to the details in our response to `Q2`.
>
> ---
> > **`Q6`:** *"[Suggestion] Consider classifier-free guidance (CFG) with semantic maps during inference."*
>
> **A:** Thank you for this thoughtful suggestion. We conducted a comprehensive study by applying CFG in our conditional steps with varying guidance weights $w \in \{ 1, 2, 4, 8 \}$, using the formula:
> $$
> \hat{\epsilon} = (1 + w)\epsilon_{\text{cond}} - w\epsilon_{\text{uncond}},
> $$
> where:
> - $\hat{\epsilon}$: final noise prediction,
> - $\epsilon_{\text{cond}}$: noise predicted with conditional input,
> - $\epsilon_{\text{uncond}}$: noise predicted with unconditional input,
> - $w$: guidance strength controlling the influence of the condition.
>
> In the conditional steps of Spiral, we set the guidance weight $ w = 0 $, which makes the inference to standard conditional generation.
>
> The results (see table below) show **minimal improvements** when $w = 1$, particularly in Cartesian metrics, and **degradation** for larger weights. We attribute this to **intermediate semantic predictions still containing artifacts**, which can mislead the generator when overly trusted. Thus, while promising in principle, CFG offers **limited gains** in our current setup. As suggested, we have included these results and the discussion in the revised supplementary material.
>
> |  | Range View | Range View | Range View | Range View | Cartesian | Cartesian | Cartesian | Cartesian | BEV | BEV | BEV | BEV |
> |:---:|:---:|:---:|:---:|:---:|:---:|:---:|:---:|:---:|:---:|:---:|:---:|:---:|
> | $w$ | FRD (x$1$) | MMD (x$10^{-2}$) | S-FRD (x$1$) | S-MMD (x$10^{-2}$) | FRD (x$1$) | MMD (x$10^{-2}$) | S-FRD (x$1$) | S-MMD (x$10^{-2}$) | JSD (x$10^{-2}$) | MMD (x$10^{-3}$) | S-JSD (x$10^{-2}$) | S-MMD (x$10^{-3}$) |
> | 0 | 170.18 | 4.81 | 382.87 | 4.10 | 8.06 | 1.10 | 153.61 | 3.20 | 3.76 | 0.15 | 9.16 | 1.41 |
> | 1 | 170.19 | 5.02 | 382.87 | 4.15 | 8.03 | 1.01 | 153.68 | 3.27 | 3.91 | 0.16 | 9.20 | 1.42 |
> | 2 | 170.30 | 5.07 | 383.71 | 4.24 | 8.48 | 1.27 | 155.98 | 3.80 | 4.02 | 0.17 | 9.25 | 1.43 |
> | 4 | 170.91 | 5.18 | 385.97 | 4.43 | 13.63 | 3.37 | 168.30 | 5.99 | 4.45 | 0.19 | 9.41 | 1.45 |
> | 8 | 173.98 | 5.42 | 391.68 | 4.61 | 26.86 | 9.45 | 195.15 | 11.04 | 5.15 | 0.23 | 9.81 | 1.54 |
>
> > ***Note:** All metrics are lower-is-better.*
>
> ---
> Last but not least, we would like to sincerely thank Reviewer `oNZH` again for the valuable time and constructive feedback provided during this review.

---

> ### Author Response · Authors · 2025-08-03
> **Looking forward to hearing from you**
>
> **Dear Reviewer `oNZH`,**
>
> Thank you once again for your thoughtful and constructive feedback!
>
> In response to your comments, we have added quantitative statistics on the closed-loop switching step, showing that it typically occurs in the final 40% of the generation process, balancing stability and influence. We also clarified the generation time of Spiral (5.7s/scene), which remains efficient for large-scale data generation. Your concern about early or late switching is well-taken and reflected in our experiments (Table 4), supporting our dynamic threshold design. As requested, we clarified the source of ground-truth semantic maps in Fig. 4 and included empirical results on applying classifier-free guidance (CFG), which yielded limited gains due to intermediate prediction noise. These updates are now included in the revised manuscript and supplementary material.
>
> We are actively participating in the Author-Reviewer Discussion phase and would be happy to provide further clarification if needed!
>
> *Best regards,*
>
> The Authors of Submission 1304

---

> ### Comment · Reviewer_oNZH · 2025-08-06
>
> Thank you for the detailed rebuttal. Regarding the answer to Q4, what confused me is how the generation model could have ground-truth segmentation annotations when the generated data is inherently unpredictable. Please correct me if I’ve misunderstood anything.

---

> ### Author Response · Authors · 2025-08-06
> **Follow-Up Response to Reviewer oNZH**
>
> **Dear Reviewer `oNZH`,**
>
> We sincerely thank you for your continued engagement!
>
> In our previous reply to `Q4`, we *misunderstood* your original question. To clarify:
> 1. The segmentation maps attached to the generated LiDAR samples are generated **jointly** with the point clouds during generation.
> 2. However, when we report evaluation results, the ground truth always refers to all **real LiDAR scenes** and their corresponding human-annotated **segmentation labels**.
>
> Below, we would like to explain this in more detail:
> - We would like to emphasize that, in generation tasks, the evaluation does **not** compare each generated sample against a specific *"ground truth"* counterpart. Instead, it evaluates the **statistical distributional similarity** between the generated set and the real set, as is commonly done with metrics like FID [`R1`] in the 2D image generation community.
>
> - Specifically:
>   - Given a generated set consisting of $M$ LiDAR scenes \{$(x_1^g, y_1^g), (x_2^g, y_2^g), ..., (x_M^g, y_M^g)$\}, where $x^g$ denotes the generated LiDAR point coordinates and $y^g$ denotes the generated semantic maps, and a real set consisting of $N$ LiDAR scenes and their human-annotated segmentation labels: \{$(x_1^r, y_1^r), (x_2^r, y_2^r), ..., (x_N^r, y_N^r)$\}. For a **generated LiDAR scene** $x_i^g$, there is **no corresponding human-annotated segmentation label**, so the evaluation cannot rely on per-sample comparisons.
>   - Instead, we transform $(x_i^g, y_i^g)$ into a **semantic-aware feature** $f^s_g$ by concatenating the encoded features from the RangeNet++ encoder and the LiDM semantic conditional module, as illustrated in Eq. `9` and Fig. `3` of the manuscript.
>   - This gives us a feature set for the **generated samples**:  $\mathcal{F}^s_g =$ \{$f^s_{g1}, f^s_{g2}, ..., f^s_{gM}$\}.
>   - Similarly, we obtain the feature set for **real samples**:  $\mathcal{F}^s_r =$ \{$f^s_{r1}, f^s_{r2}, ..., f^s_{rN}$\}.
>   - We then compute the statistical similarity between $\mathcal{F}^s_g$ and $\mathcal{F}^s_r$ using our proposed metrics, such as **S-FRD** and **S-MMD**, as defined in Eqs. `2`-`5` in the supplementary material.
>
> - Only when the generated LiDAR scenes exhibit both **high geometric quality** and **strong geometric-semantic alignment** (i.e., well-aligned semantic maps), will the features $f^s_g$ fall into a reasonable domain and $\mathcal{F}^s_g$ share statistical properties with $\mathcal{F}^s_r$.
>
> - Thus, by comparing the **semantic-aware feature distributions**, we bypass the need for ground-truth semantic labels for the generated samples.
>
> We appreciate your thoughtful question and hope the above explanation resolves the ambiguity. We are happy to elaborate further if any details are still unclear.
>
> *Yours sincerely,*
>
> The Authors of Submission 1304
>
> ---
> **Reference:**
>
> - [`R1`] Heusel, Martin, et al. *"GANs Trained by a Two Time-Scale Update Rule Converge to a Local Nash Equilibrium."* Advances in Neural Information Processing Systems 30, 2017.

---

> > ### Comment · Reviewer_oNZH · 2025-08-06
> >
> > Thank you for the detailed explanation.
> > I have no further questions.
> >
> > Although the fact that the proposed method is approximately 2.05 seconds slower than the two-stage baseline (R2DM + RangeNet++) is a minor drawback for me, the authors have addressed the remaining concerns—particularly the statistical analysis on the denoising timestep.
> >
> > Accordingly, I will retain my score as Borderline Accept.

---

### Official Review · Reviewer_gApr · 2025-07-02

**Clarity:** 3
**Significance:** 2
**Originality:** 2
**Rating:** 4
**Confidence:** 3

**Summary:**

The authors propose SPIRAL, a LiDAR diffusion model that jointly generates depth, reflectance, and pixel-wise semantic labels from noise. They tune the diffusion model to be conditioned on the semantic map as well as outputting the map, thus yielding a closed-loop in terms of both training and inference. They also propose several new metrics for semantic alignment evaluation of the LIDAR generative models. Through the experiment, they show their method are superior compared to their baselines, which is previous LIDAR generative models with a semantic prediction network.

**Questions:**

- What is the memory and runtime comparing to your baselines?
- The proposed S-x metrics compares the semantic map with the ones in the dataset, but **it does not account for the alignment between the generated scene and the map**. For example, say one model generates a garbage scene, but the segmentor segments the scene perfectly; in this case, the metrics would be double-dropped. In other words, the scene generation quality is overly out-weighted to the segmentation quality in such measurement.
- Closed/Open loop inference: if I understand it correctly, the open loop is directly diffusing the semantic map and LIDAR. Intuitively, I think this would cause more problems in terms of discrepancy between the two modalities. The metrics, however, are not able to reflect if the map is aligned with the generated scene.
- The Data Augmentation results should be more extensively compared, as this is one of the main applications mentioned in the paper
    - It would be great if the authors show the mIOU of SPVCNN++ without any augmented data for comparison
    - The author listed a column of “None” in Table C but failed to explain this column.
    - Will the result still be significant in ratios where the real data is abundant, such as 80%?

**Ethical Concerns:**

["NO or VERY MINOR ethics concerns only"]

**Final Justification:**

I think the proposed data synthetic method makes sense, and in the rebuttal, the authors addressed my concern about runtime and applications. Unlike other reviewers, I don't think segmentation is utterly insignificant, and the author did show that the synthetic data improves the models, which could be a good application.

On the other hand, the improvement is still quite marginal, and only a percentage is provided. This could potentially be an indication of the diversity of synthesized data. The paper could benefit from a wider range of experiment on data augmentation.

Thus, I remain my score as 4, weakly accept

**Limitations:**

The authors included limitations and societal impact in the appendix.

See questions for weakness.

**Quality:**

3

**Strengths And Weaknesses:**

* The closed-loop training/inference scheme of diffusion models is novel and reasonable.
* Generating Semantic information along with the LIDAR makes sense for achieving a better and more precise alignment
* The experiments show that their method passes the baseline in both scene and semantic map quality

---

> ### Author Rebuttal · Authors · 2025-07-29
>
> We sincerely thank Reviewer `gApr` for the thoughtful and constructive feedback. We are encouraged that you found our closed-loop inference design both “novel” and “reasonable,” and appreciated our improvements in semantic-scene alignment. Below, we address each of your concerns in detail.
>
> ---
> > **`Q1`:** *"What is the memory and runtime compared to your baselines?"*
>
> **A:** Thank you for this practical question. The parameter sizes of Spiral and baselines are reported in the “Param” column of Table `1` and Table `2` in the manuscript. The **average inference time** per LiDAR scene is presented in Table `B` and Section `F` of the supplementary material. Specifically, Spiral takes **5.7 seconds per scene** on an NVIDIA A6000 GPU, which is **comparable to or faster** than two-stage baselines, thanks to its streamlined end-to-end generation pipeline.
>
> ---
> > **`Q2`:** *"The proposed S-x metrics compare the semantic map with the ones in the dataset, but it does not account for the alignment between the generated scene and the map. For example, a garbage scene with perfect segmentation would still pass the metric. This might over-weight scene generation in the evaluation."*
>
> **A:** We appreciate this insightful concern. In practice, we find that the scenario of a *“garbage scene”* with strong semantic segmentation performance **rarely occurs**. From both empirical observation and theoretical reasoning:
> - In our two-stage baselines (e.g., LiDARGen + RangeNet++), **moderately corrupted** LiDAR scenes already lead to significant segmentation failures, even when the segmentation network is extensively trained with noise augmentations. This is expected, since **3D segmentation is fundamentally geometry-dependent** — poorly generated structures yield unreliable spatial features and semantic predictions.
> - Our **semantic-aware evaluation metrics** are explicitly designed to measure cross-modal alignment, not scene quality in isolation. As illustrated in Fig. `3`(a), we encode the depth image and semantic map **independently** using two modality-specific encoders (RangeNet++ and LiDM-Seg), and fuse the extracted features into a **unified semantic-aware feature** $f_s$. Therefore, even in the rare cases where a scene may have poor geometric quality but good semantic map, the geometric quality is **not** *"overly out-weighted"*.
> - At the BEV level (please see Fig. `3`(b)), we compute **semantic histograms** across classes and spatial cells. These are highly sensitive to both the **correct geometry** (e.g., point distribution) and the **correct semantic allocation** (e.g., class labels). If either modality fails, due to noisy generation or poor segmentation, the BEV histogram will diverge from the real data, and the model will be penalized accordingly.
>
> In summary, geometry and semantics contribute equally in our metrics by design, and misalignment across modalities is indeed captured. As suggested, we have further clarified this detail in the revised manuscript.
>
> ---
> > **`Q3`:** *"Regarding closed/open-loop inference: I understand that open-loop directly diffuses both modalities, which might lead to discrepancies. However, the metrics may not be able to reflect misalignment between the generated scene and the map."*
>
> **A:** Your understanding of the open-loop setting is correct — it involves jointly generating the LiDAR scene and semantic map **without** enforcing consistency during generation. This indeed introduces **cross-modality discrepancies**.
>
> - As shown in Table `4`, closed-loop inference with default hyperparameters (the highlighted row) outperforms open-loop inference (first row). Importantly, we emphasize that the semantic map in closed-loop inference is still generated by Spiral, and is **not** externally provided. Therefore, we believe that the proposed closed-loop inference strategy is a novel contribution of this work.
> - Our proposed semantic-aware metrics are designed to effectively evaluate **cross-modality alignment**.
>   - For **range-view** and **Cartesian-based metrics**, as illustrated in Fig. `3`(a), the semantic-aware feature $f_s$ for each scene is formed by concatenating the global features extracted from the depth image and semantic map, respectively. This ensures that both geometric and semantic quality are jointly evaluated.
>   - For **BEV-based metrics**, they assess alignment at a **finer** granularity. Specifically, for each generated scene, we compute semantic-aware histograms $h_s$ over the bird’s-eye-view by combining the histogram of each semantic class (please see Fig. `3`(b)). As a result, a generated scene with either:
>     1. good semantic segmentation but poor geometric quality, or;
>     2. poor semantic segmentation but good geometric quality
>
> will fail to produce a histogram that matches the real scene. Additionally, this also ensures that the BEV metrics penalize the misalignment in geometry or semantics, making them strong indicators of overall generation fidelity.
>
> Furthermore, as elaborated in `Q2`, our semantic-aware metrics, including modality-specific encoders and BEV histograms, are designed to detect cross-modal misalignment, ensuring that **semantic and structural coherence** is thoroughly assessed.
>
> ---
> > **`Q4.1`:** *"The data augmentation results should be more extensively compared, especially as this is a main application of the paper."*
>
> **A:** We appreciate your emphasis on this point. In addition to the experiments presented in the manuscript, we further conduct a comprehensive evaluation of generative data augmentation (GDA) under two challenging tests:
>
> 1. **Cross-Domain Generalization (Waymo Test):**
>     - We use LiDAR scenes generated by Spiral, trained on SemanticKITTI, to augment the training set of Waymo [`R1`] (when 10% of real data is available), and evaluate the model's performance on the Waymo val set.
>     - The segmentation model SPVCNN++ is trained under three settings:
>       1. without GDA,
>       2. with synthetic data generated by R2DM & RangeNet++,
>       3. with our Spiral-generated data.
>     - The segmentation performance (mIoU) under these settings is **55.89%**, **55.93%**, and **57.52%**, respectively. These results demonstrate the **strong generalization ability** of our Spiral-based GDA approach.
>
> 2. **Out-of-Distribution (OOD) Extreme Weather (Robo3D Test):**
>     - We evaluate SPVCNN++ trained under the same three settings as in the above "Cross-Domain Generalization Test", on challenging out-of-distribution subsets, `fog` and `wet ground`, from *Robo3D* [`R2`].
>     - The results demonstrate that, although Spiral is not specifically fine-tuned for these OOD extreme weather conditions, the data generated by Spiral still **effectively enriches the training set** and **improves performance**.
>
> > ***Table.** Experiments under `fog` scenarios:*
> | Ratio of Real Data | w/o GDA | R2DM & RangeNet++ | Spiral (Ours) |
> |:---:|:---:|:---:|:---:|
> | 1% | 32.07 | 36.82 | **44.06** |
> | 10% | 53.93 | 54.20 | **58.61** |
> | 20% | 55.89 | 56.07 | **61.24** |
>
> > ***Table.** Experiments under `wet ground` scenarios:*
> | Ratio of Real Data | w/o GDA | R2DM & RangeNet++ | Spiral (Ours) |
> |:---:|:---:|:---:|:---:|
> | 1% | 34.26 | 37.96 | **44.19** |
> | 10% | 54.70 | 55.92 | **59.31** |
> | 20% | 56.81 | 57.53 | **62.02** |
>
> Across all scenarios, our Spiral-based GDA consistently outperforms all baselines. These results demonstrate our **effectiveness in both cross-domain and OOD generalization settings**. As suggested, we have improved the presentation of these results and emphasized the comparisons further in the revised manuscript.
>
> ---
> > **`Q4.2`:** *"Please show the mIoU of SPVCNN++ without any augmented data."*
>
> **A:** Thank you for pointing this out. The mIoU results of SPVCNN++ trained **without** any generative data augmentation (GDA) are already reported in the “None” column of Table `C` in the supplementary material (see also "w/o" column of Table `3` in the manuscript).
>
> This baseline reflects the model trained using only real data. For example, under the 10% supervision setting on the Robo3D-fog set, this baseline achieves **53.93%** mIoU, while Spiral-based GDA improves the score to **58.61%**. We have revised the table caption to make this clearer.
>
> ---
> > **`Q4.3`:** *"The 'None' column in Table C is unclear and not explained."*
>
> **A:** Thanks for pointing out our oversight! The “None” column in Table `C` (corresponding to the "w/o" column in Table `3` in the main paper) refers to the baseline performance of SPVCNN++ **trained without any synthetic data**, i.e., **real-only training**. This column is essential for measuring the absolute contribution of generative augmentation. We have revised the table caption and corresponding text to explicitly define “None” and "w/o" to avoid confusion.
>
> ---
> > **`Q4.4`:** *"What happens when the real data ratio is high (e.g., 80%)?"*
>
> **A:** Great question. We have conducted additional experiments under the **80% supervision setting**, where the amount of real data is abundant. The resulting mIoU scores on the SemanticKITTI are:
> - None (real only): 64.48%
> - R2DM + RangeNet++: 64.51%
> - Spiral (ours): **65.44%**
>
> Even in this data-rich regime, Spiral continues to provide **non-trivial performance gains**, while R2DM shows negligible benefit. This result highlights that Spiral not only excels in low-data regimes but also contributes value in high-data scenarios by generating **semantically aligned and diverse** LiDAR scenes.
>
> ---
> **References:**
> - [`R1`] Sun, Pei, et al. "Scalability in Perception for Autonomous Driving: Waymo Open Dataset." CVPR, 2020.
> - [`R2`] Kong, Lingdong, et al. "Robo3D: Towards Robust and Reliable 3D Perception against Corruptions." ICCV, 2023.
> ---
> Last but not least, we would like to sincerely thank Reviewer `gApr` again for the valuable time and constructive feedback provided during this review.

---

> > ### Comment · Reviewer_gApr · 2025-08-06
> >
> > Thank you for the detailed rebuttal. Your clarifications resolve my concerns. I'd recommend adding these to the final version

---

> > > ### Author Response · Authors · 2025-08-06
> > > **Thank you for your acknowledgment**
> > >
> > > **Dear Reviewer `gApr`,**
> > >
> > > We sincerely appreciate your acknowledgment of our rebuttal. We are glad to hear that your questions have been addressed and are voting toward acceptance.
> > >
> > > We commit to incorporating the clarifications and insights discussed to ensure better clarity and completeness.
> > >
> > > Your support and encouragement mean a lot to us!
> > >
> > > *Best regards,*
> > >
> > > The Authors of Submission 1304

---

> ### Author Response · Authors · 2025-08-03
> **Looking forward to hearing from you**
>
> **Dear Reviewer `gApr`,**
>
> Thank you once again for your thoughtful and constructive review!
>
> In response to your comments, we clarified the runtime and memory usage of Spiral compared to baselines, and explained how our semantic-aware metrics jointly assess geometric and semantic fidelity to capture modality misalignment. We also emphasized the role of closed-loop inference in promoting semantic-scene consistency and provided further evaluation results under both cross-domain (Waymo) and out-of-distribution (Robo3D) settings. These results demonstrate consistent performance gains over existing augmentation methods, even under high-data regimes. We have revised the manuscript and supplementary materials accordingly to improve clarity and completeness.
>
> We are actively participating in the Author-Reviewer Discussion phase and would be happy to provide further clarification if needed!
>
> *Best regards,*
>
> The Authors of Submission 1304

---

### Official Review · Reviewer_XYPz · 2025-07-03

**Clarity:** 3
**Significance:** 3
**Originality:** 2
**Rating:** 3
**Confidence:** 4

**Summary:**

The paper introduces SPIRAL, a semantic-aware range-view LiDAR diffusion model for jointly generating semantic labels, depth, and reflectance images. The authors further extend evaluation metrics with semantic awareness to comprehensively assess the generated LiDAR scenes, showing good results across the benchmarks.

**Questions:**

All of my concerns are listed in the weaknesses section, and I may adjust the rating if they are well addressed.

**Ethical Concerns:**

["NO or VERY MINOR ethics concerns only"]

**Final Justification:**

I would like to thank the authors for addressing my questions point-by-point. While the rebuttal resolves some of my concerns, I still have the concern regarding the practical utility of the generated data. Therefore, I am inclined to maintain my original score.

**Limitations:**

Yes.

**Paper Formatting Concerns:**

None.

**Quality:**

2

**Strengths And Weaknesses:**

Strengths:
1) Simultaneously generating LiDAR scenes with semantic labels is a useful capability for the community, as it enables the creation of synthetic data for training and evaluating perception models.
2) Semantic-aware metrics are introduced for a comprehensive evaluation, and the numerical results are good when compared to baselines.
3) The paper is well-written, the problem is clearly specified and the solution is well presented.

Weaknesses:
1) Limited semantic label set: Semantic output is constrained to pre-defined categories from SemanticKITTI and nuScenes, reducing applicability to varied domains.
2) Additionally, long-tailed label distributions in LiDAR data (e.g., rare classes like bicyclists) can cause the model to underfit these classes. The paper doesn’t detail any strategies (e.g., re-weighting) to balance this.
3) Although the approach reduces collection & annotation costs, since the model is trained on a specific dataset, it is questionable how well it captures real-world variability and complexity, which may limit the practical utility of the generated data.

---

> ### Author Rebuttal · Authors · 2025-07-28
>
> We sincerely thank Reviewer `XYPz` for the thoughtful and constructive feedback. We are encouraged that you found our work “useful,” “well-written”, and that our results demonstrate “good” performance. Below, we address your key concerns in detail.
>
> ---
> >**`Q1`:** *"Limited semantic label set: semantic output is constrained to pre-defined categories from SemanticKITTI and nuScenes, reducing applicability to varied domains."*
>
> **A:** Thanks for your question. Our current design follows a widely accepted convention in 3D perception benchmarks, where tasks such as segmentation and detection rely on **fixed taxonomies**, as in SemanticKITTI and nuScenes. These datasets are **representative and diverse**, encompassing a broad range of scenes (urban, residential, highway, rural) across different geographic regions (Germany, Singapore, USA), and have become de facto standards in the field.
>
> While constrained to pre-defined classes, these benchmarks include out-of-distribution (OOD) labels, such as the “unlabeled” category in SemanticKITTI (e.g., tunnel walls, scaffolding, bus stops). Spiral is capable of modeling such categories as well, ensuring that OOD content is not omitted in our generative process.
>
> We believe that extending Spiral to support an unlimited number of categories is a **meaningful direction**. However, it faces a critical bottleneck: the **lack of reliable text–LiDAR paired data**, as also observed by *LiDM* [`R1`] and *Text2LiDAR* [`R2`].
> Although *Text2LiDAR* [`R2`] introduces language annotations, they are typically **coarse, scene-level descriptions** (e.g., *"cars and buses at a busy street"*, *"surrounded by cars"*) and lack **fine-grained, point-level semantics** that are essential for point-wise labeled LiDAR scene generation. This makes open-vocabulary generation technically under-constrained. Addressing this challenge is a key goal in our ongoing research. Thanks again for your valuable suggestion.
>
> ---
> > **`Q2`:** *"Long-tailed label distributions in LiDAR data (e.g., rare classes like bicyclists) can cause the model to underfit these classes. The paper doesn’t detail any strategies (e.g., re-weighting) to balance this."*
>
> **A:** Thank you for highlighting this. We acknowledge the class imbalance present in both SemanticKITTI and nuScenes. To address this, Spiral incorporates a **class-reweighting** strategy, consistent with the official RangeNet++ codebase.
>
> Specifically, the loss weight for category $i$ is defined as:
> $$
> w_i = \frac{1}{\text{freq}_i + \epsilon},
> $$
> where $\text{freq}_i$ is the normalized frequency of category $i$, and $\epsilon$ is a small constant (default: $0.001$).
>
> This strategy mitigates the dominance of frequent classes and encourages the model to attend to **under-represented semantic categories** during training. As suggested, we have further clarified this detail in the revised manuscript.
>
> ---
> > **`Q3`:** *"Although the approach reduces collection and annotation costs, since the model is trained on a specific dataset, it is questionable how well it captures real-world variability and complexity, which may limit the practical utility of the generated data."*
>
> **A:** We appreciate this concern. To evaluate how well Spiral captures **real-world variability** and how robust it is to **real-world complexity**, we conduct experiments on generative data augmentation (GDA) under **two challenging scenarios**:
> 1. **Cross-Domain Generalization (Waymo Test):**
>     - We use LiDAR scenes generated by Spiral, trained on SemanticKITTI, to augment the training set of Waymo [`R4`] (when 10% of real data is available), and evaluate the model's performance on the Waymo val set.
>     - The segmentation model SPVCNN++ is trained under three settings:
>       1. without GDA,
>       2. with synthetic data generated by R2DM & RangeNet++,
>       3. with our Spiral-generated data.
>     - The segmentation performance (mIoU) under these settings is **55.89%**, **55.93%**, and **57.52%**, respectively. These results demonstrate the **strong generalization ability** of our Spiral-based GDA approach.
>
> 2. **Out-of-Distribution (OOD) Extreme Weather (Robo3D Test):**
>     - We evaluate SPVCNN++ trained under the same three settings as in the above "Cross-Domain Generalization Test", on challenging out-of-distribution subsets, `fog` and `wet ground`, from *Robo3D* [`R3`].
>     - The results demonstrate that, although Spiral is not specifically fine-tuned for these OOD extreme weather conditions, the data generated by Spiral still **effectively enriches the training set** and **improves performance** under these challenging scenarios.
>
> > ***Table.** Experiments under `fog` scenarios:*
> | Ratio of Real Data | w/o GDA | R2DM & RangeNet++ | Spiral (Ours) |
> |:---:|:---:|:---:|:---:|
> | 1% | 32.07 | 36.82 | **44.06** |
> | 10% | 53.93 | 54.20 | **58.61** |
> | 20% | 55.89 | 56.07 | **61.24** |
>
> > ***Table.** Experiments under `wet ground` scenarios:*
> | Ratio of Real Data | w/o GDA | R2DM & RangeNet++ | Spiral (Ours) |
> |:---:|:---:|:---:|:---:|
> | 1% | 34.26 | 37.96 | **44.19** |
> | 10% | 54.70 | 55.92 | **59.31** |
> | 20% | 56.81 | 57.53 | **62.02** |
>
> These results confirm that Spiral generates **semantically and structurally diverse scenes** that enhance real-world performance under both domain and weather shifts — supporting its practical utility.
>
> ---
> **References:**
> - [`R1`] Ran, Haoxi, et al. "Towards Realistic Scene Generation with LiDAR Diffusion Models." Proceedings of the IEEE/CVF Conference on Computer Vision and Pattern Recognition, 2024.
> - [`R2`] Wu, Yang, et al. "Text2LiDAR: Text-Guided LiDAR Point Cloud Generation via Equirectangular Transformer." European Conference on Computer Vision, 2024.
> - [`R3`] Kong, Lingdong, et al. "Robo3D: Towards Robust and Reliable 3D Perception against Corruptions." Proceedings of the IEEE/CVF International Conference on Computer Vision, 2023.
> - [`R4`] Sun, Pei, et al. "Scalability in Perception for Autonomous Driving: Waymo Open Dataset." Proceedings of the IEEE/CVF Conference on Computer Vision and Pattern Recognition, 2020.
>
> ---
> Last but not least, we would like to sincerely thank Reviewer `XYPz` again for the valuable time and constructive feedback provided during this review.

---

> ### Author Response · Authors · 2025-08-03
> **Looking forward to hearing from you**
>
> **Dear Reviewer `XYPz`,**
>
> Thank you once again for your thoughtful and constructive feedback!
>
> In response to your comments, we clarified the scope and motivation of using fixed semantic taxonomies (e.g., SemanticKITTI, nuScenes), and discussed the potential and current limitations of open-vocabulary generation due to the scarcity of fine-grained text–LiDAR pairs. We also detailed the class-reweighting strategy adopted during training to mitigate long-tailed label distributions. Finally, we added new results on cross-domain (Waymo) and OOD (Robo3D) scenarios, demonstrating that Spiral-generated data significantly improves real-world robustness and generalization.
>
> Your suggestions have greatly helped us improve the clarity and completeness of our work.
>
> We are actively participating in the Author-Reviewer Discussion phase and would be happy to provide further clarification if needed!
>
> *Best regards,*
>
> The Authors of Submission 1304

---

> > ### Comment · Reviewer_XYPz · 2025-08-08
> > **Official Comment by Reviewer XYPz**
> >
> > Dear authors, thank you for addressing my questions point-by-point. Your rebuttal resolves some of my concerns. However, I still have the concern regarding the practical utility of the generated data.  Given that the main contribution of the work is to generate synthetic data for augmentation purposes, I find the setup - training directly on synthetic datasets with a limited semantic label set - somewhat restrictive for this goal. Overall, I am inclined to maintain my original score.

---

> > > ### Author Response · Authors · 2025-08-08
> > > **Clarification on Spiral’s Core Contributions and Scope**
> > >
> > > Dear Reviewer `XYPz`,
> > >
> > > Thank you for taking the time to consider our responses!
> > >
> > > We would like to emphasize that our paper’s primary focus is **generation**, rather than **segmentation**, as stated in our *General Response*. The first three key contributions are:
> > > 1. The **first** unified diffusion framework that jointly generates LiDAR geometry and semantic maps,
> > > 2. The **closed-loop inference** strategy of Spiral that improves the generative quality of LiDAR scenes, and
> > > 3. Novel **semantic-aware metrics** (S-FPD, S-MMD, S-JSD).
> > > The data augmentation experiment is an application test of Spiral. Prior to Spiral, such augmentation tests were **absent** due to the fidelity limitations of generative models and the lack of semantic labels for generated scenes [`R1`–`R4`].
> > >
> > > While open-vocabulary generation is indeed valuable, it is **orthogonal** to our goal of high-quality LiDAR scene generation. As we mentioned earlier, the main bottleneck in this area is the lack of suitable  text–LiDAR paired data [`R1`–`R2`].
> > >
> > > Despite being trained with a fixed set of semantic labels, Spiral still provides a **foundation** for open-vocabulary LiDAR scene understanding:
> > >
> > > - **Generative perspective:** Spiral can model out-of-distribution (OOD) objects (e.g., tunnel walls, scaffolding, bus stops in SemanticKITTI), ensuring that OOD content is retained in the generation process.
> > > - **Downstream segmentation perspective:** In the Waymo cross-domain generalization test, we demonstrated that Spiral trained on SemanticKITTI can, via category mapping, generate augmentation data for a dataset containing categories **unseen** during training.
> > >
> > > Lastly, we would like to thank Reviewer `XYPz` again for the valuable time and feedback provided during this review.
> > >
> > > *Warm regards,*
> > >
> > > The Authors of Submission 1304
> > >
> > > ---
> > >
> > > **References:**
> > > - [`R1`] Ran, Haoxi, et al. "Towards Realistic Scene Generation with LiDAR Diffusion Models." Proceedings of the IEEE/CVF Conference on Computer Vision and Pattern Recognition, 2024.
> > > - [`R2`] Wu, Yang, et al. "Text2LiDAR: Text-Guided LiDAR Point Cloud Generation via Equirectangular Transformer." European Conference on Computer Vision, 2024.
> > > - [`R3`] Zyrianov, Vlas, Xiyue Zhu, and Shenlong Wang. "Learning to generate realistic lidar point clouds." European Conference on Computer Vision. Cham: Springer Nature Switzerland, 2022.
> > > - [`R4`] Nakashima, Kazuto, and Ryo Kurazume. "Lidar data synthesis with denoising diffusion probabilistic models." 2024 IEEE International Conference on Robotics and Automation (ICRA). IEEE, 2024.

---

### Author Response · Authors · 2025-08-05
**General Response**

**Dear Reviewers, ACs, and SACs,**

We sincerely thank you for your valuable time, thoughtful feedback, and engaging discussions throughout the review process!

---

We are encouraged by the **recognition** of our contributions across multiple aspects:

* Reviewer `XYPz` found our work *"useful"*, *"well-written"*, and appreciated that our model achieves *"good performance"* while *"enabling synthetic data for training and evaluation"*.
* Reviewer `gApr` highlighted the *"novelty and reasonableness of the closed-loop scheme"*, the *"improvements in semantic-scene alignment"*, and the *"superior results compared to baselines"*.
* Reviewer `oNZH` praised our *"state-of-the-art performance"*, *"practical utility for downstream segmentation"*, and *"cross-modal consistency via closed-loop inference"*.
* Reviewer `ZJCH` recognized our *"innovative pipeline for simultaneous LiDAR and semantic generation"* and *"novel evaluation metrics (S-FPD, S-MMD)"*.

---
In response to your constructive suggestions, we have made the following **clarifications and improvements**:

* **Scope & Applicability**
  * As suggested by Reviewer `XYPz`, we clarified that while Spiral currently supports pre-defined label sets (SemanticKITTI, nuScenes), it also models OOD content and can be extended to open-vocabulary settings with more granular text–LiDAR supervision.
  * As suggested by Reviewer `ZJCH`, we emphasized that Spiral supports both unconditional and conditional generation, and explained how the unconditional setting enables cost-effective augmentation.
  * As suggested by Reviewer `ZJCH`, we clarified that all reported evaluations focus on unconditional generation, and the ground truth is drawn from standard LiDAR datasets with full annotations.

* **Closed-Loop Inference**
  * As suggested by Reviewers `oNZH` and `gApr`, we provided detailed statistics on the switching behavior (e.g., average activation step: 154 for SemanticKITTI), and explained how this timing ensures semantic stability and avoids inefficient early or late switching.
  * As suggested by Reviewer `oNZH`, we clarified the architectural distinction between Spiral’s joint generation and the two-step post-hoc methods used in baselines.

* **Evaluation Metrics & Design**
  * As suggested by Reviewer `gApr`, we elaborated on how our semantic-aware metrics jointly assess structural and semantic alignment via modality-specific encoders and BEV-level histograms.
  * As suggested by Reviewer `ZJCH`, we improved the explanation of Table 3 to clarify the setup and the meaning of the “w/o” (None) baseline across real data ratios.

* **Generative Data Augmentation**
  * As suggested by Reviewers `XYPz`, `gApr`, and `ZJCH`, we added extensive evaluations of Spiral-based data augmentation on Waymo and Robo3D benchmarks under cross-domain and OOD settings.
  * As suggested by Reviewer `gApr`, we emphasized that Spiral consistently outperforms prior methods (e.g., R2DM + RangeNet++) across low- and high-data regimes, with gains up to +11.99% in fog scenarios.

* **Efficiency & Scalability**
  * As suggested by Reviewers `gApr` and `oNZH`, we highlight the runtime (5.7s/scene on A6000) and parameter comparisons reported in the manuscript and supplementary materials, showing that Spiral is efficient and suitable for scalable generation.

* **Ablations & Design Choices**
  * As suggested by Reviewer `oNZH`, we provided empirical results on classifier-free guidance, showing limited benefits due to semantic noise and reinforcing the utility of our native closed-loop conditioning.

---
We would like to re-emphasize the **key contributions** of our work:

* Spiral, a unified diffusion framework that jointly generates LiDAR geometry and semantic maps in an end-to-end manner — advancing beyond post-hoc segmentation or two-stage designs.
* A *closed-loop inference strategy* that dynamically conditions generation on high-confidence semantic predictions, promoting cross-modal alignment and semantic faithfulness.
* New *semantic-aware metrics* (S-FPD, S-MMD, S-JSD) that effectively capture geometric–semantic consistency across modalities and granularities (range, Cartesian, BEV).
* Strong empirical results demonstrating that Spiral-generated data improves downstream segmentation performance under *domain shift*, *weather corruptions*, and *low-data regimes*, with competitive runtime and model efficiency.

---
With **two days** remaining in the **Author-Reviewer Discussion** phase (*August 6, Anywhere on Earth*), we welcome any further feedback and are happy to continue the discussion.

Thank you once again for your thoughtful reviews and kind support!

*Warmest regards,*

The Authors of Submission 1304

---

### Note · Authors · 2025-08-14

**Dear Reviewers, ACs, and SACs,**

We sincerely appreciate the opportunity to provide a final remark. In our earlier **General Response**, we summarized Spiral's key contributions, reviewer acknowledgements, and clarifications provided during the review process. Here, we offer a summary of the reviewer-author discussion phase.

We are pleased that reviewers `gApr` and `oNZH` have their concerns fully addressed and now support the acceptance of Spiral. We will incorporate the clarifications into the manuscript.

Reviewers `XYPz` and `ZJCH` each provided one comment. Although we responded in detail and invited further discussion, **no** follow-up feedback was received. We are confident that our responses adequately address their concerns. Below, we summarize these comments and our replies, and we respectfully recommend that ACs and SACs refer to the original responses if time permits.

- **Regarding reviewer XYPz’s concern**: *"Practical utility of the generated data due to the limited semantic label set"*
    - We follow the **standard** taxonomies of **SemanticKITTI** and **nuScenes**, the most widely used benchmarks in autonomous driving covering 3 continents and diverse road conditions.
    - Spiral-generated data has demonstrated utility in **extreme weather scenarios** (**Robo3D**) and on **unseen categories** (**Waymo**), showing strong generalization.
    - Spiral explicitly models long-tail OOD objects instead of ignoring them.
    - While open-vocabulary research is valuable, it is **orthogonal** to our focus: generating high-quality LiDAR scenes rather than segmentation. Spiral achieves SOTA performance in both geometric and semantic-aware evaluations.


- **Regarding reviewer ZJCH’s comment**: *"Experiment setup using all real data"*
    - We conducted this experiment on **SemanticKITTI** and **Robo3D**, and the results confirm the effectiveness of Spiral-generated data.

- **Regarding reviewer ZJCH’s comment**: *"Several works have already achieved LiDAR generation conditioned on bounding boxes or keypoints"*
    - The three works cited by `ZJCH` do **not** rely solely on bounding boxes or keypoints; they use richer input information. To our knowledge, **no existing work** can generate a complete LiDAR scene using only bounding boxes or keypoints.


We once again thank the reviewers, ACs and SACs for your valuable time in handling our submission.

*With sincere thanks,*
The Authors of Submission 1304

---

### Decision · Program_Chairs · 2025-09-17

**Decision:**

Accept (poster)

**Comment:**

The paper presents a diffusion model for generating depth, reflectance images, and semantic images from range-view lidar data. Reviewers appreciated that the paper is well-written, the model achieves good performance vs. baselines, and the closed-loop training and inference schemes are novel and reasonable. They also highlighted the following weaknesses of the approach.
- The paper does not assess when the model switches between unconditional and conditional generation (to incorporate the semantic information).
- The evaluation lacks an assessment of inference speed.
- The model does not handle temporal generation of lidar data.
- The semantic output is limited to pre-defined categories from existing datasets.
- The method may not generalize to rare classes (e.g., bicyclists) or work well on real-world data since it is trained on a specific dataset.

The authors provided a rebuttal to respond to the concerns — they included a new analysis to assess when the model switches to conditional generation, an evaluation of inference speed, an assessment of cross-domain generalization and out-of-distribution performance, as well as a number of other clarifications. In response to the rebuttal, one reviewer raised their score (from Reject to Borderline Reject), and the final reviewer ratings remained split with two ratings of Borderline Accept and two ratings of Borderline Reject.

After reading the paper, I'm persuaded by the remarks of the positive reviewers. In particular, I found the evaluation of the method to be thorough, and the technique does appear to be the state-of-the-art for the task of lidar scene generation, with significant improvements in the consistency of the generated semantic labels with the ground truth datasets compared to baselines. Additionally, the authors appear to have responded to most concerns in the very thorough rebuttal, and including the new results will further improve the paper. Overall, while the paper does not provide significantly new capabilities (e.g., temporal generation), it does provide technical contributions that are potentially valuable to the community as well as improved results on an established problem. I recommend acceptance.